# Recent Advancements in Polyphenylsulfone Membrane Modification Methods for Separation Applications

**DOI:** 10.3390/membranes12020247

**Published:** 2022-02-21

**Authors:** Arun Kumar Shukla, Javed Alam, Mansour Alhoshan

**Affiliations:** 1King Abdullah Institute for Nanotechnology, King Saud University, P.O. Box 2455, Riyadh 11451, Saudi Arabia; ashukla@ksu.edu.sa; 2Department of Chemical Engineering, College of Engineering, King Saud University, P.O. Box 2455, Riyadh 11451, Saudi Arabia; 3K.A. CARE Energy Research and Innovation Center at Riyadh, P.O. Box 2022, Riyadh 11451, Saudi Arabia

**Keywords:** membranes, modification methods, polyphenylsulfone, separation

## Abstract

Polyphenylsulfone (PPSU) membranes are of fundamental importance for many applications such as water treatment, gas separation, energy, electronics, and biomedicine, due to their low cost, controlled crystallinity, chemical, thermal, and mechanical stability. Numerous research studies have shown that modifying surface properties of PPSU membranes influences their stability and functionality. Therefore, the modification of the PPSU membrane surface is a pressing issue for both research and industrial communities. In this review, various surface modification methods and processes along with their mechanisms and performance are considered starting from 2002. There are three main approaches to the modification of PPSU membranes. The first one is bulk modifications, and it includes functional groups inclusion via sulfonation, amination, and chloromethylation. The second is blending with polymer (for instance, blending nanomaterials and biopolymers). Finally, the third one deals with physical and chemical surface modifications. Obviously, each method has its own limitations and advantages that are outlined below. Generally speaking, modified PPSU membranes demonstrate improved physical and chemical properties and enhanced performance. The advancements in PPSU modification have opened the door for the advance of membrane technology and multiple prospective applications.

## 1. Introduction

Polyphenylsulfone (PPSU) belongs to sulfone-family polymers that have been thoroughly studied for their potential applications in membrane science and technology. The state-of-the-art PPSU-based membranes show superior properties, including excellent thermal and mechanical stability, high chemical resistance, impact resistance, and hydrolytic stability [1,2,3,4,5,6,7,8]. This stability can be attributed to the difference in their backbone structure compared to other polymeric materials. However, the membrane morphological structure and properties are elaborated by the composition and operating conditions of a PPSU solution, including concentration, additives, solvent type, temperature, kinetic factors, the coagulation bath (phase inversion process), etc. [9,10,11,12,13,14,15,16]. Thermodynamics and kinetics play significant roles during membrane development [17,18,19]. Thermodynamics determines whether a PPSU polymer solution is stable or not. Kinetics plays a key role in the phase separation speed. Despite the aforementioned important properties of PPSU polymers, there has been a limited number of studies concerning the preparation of PPSU membranes [20,21,22,23]. However, these studies showed promising results in polymer applications. In particular, the membranes can be used for ultrafiltration, nanofiltration, and reverse and forward osmosis. At the same time, other polymer materials often prove more susceptible in terms of stability (chemical, thermal, and mechanical) and are often expensive [24,25,26,27,28,29,30].

One disadvantage of PPSU-based membranes is their hydrophobic nature, which leads to reduced surface energy. The latter causes poor antifouling ability by foulant pollutants in water. Two more disadvantages of the PPSU membrane are its low water permeability and high fouling ability. These two have limited its application in aqueous phase separation. A number of studies have concluded that membrane fouling is directly related to hydrophobicity and surface charge, as reviewed by several researchers, while the opposite has also been reported [31,32,33,34]. Membrane fouling is generally classified as organic fouling, inorganic fouling, or biofouling (nonpolar solutes, hydrophobic particles, microorganism, mineral scale). It can easily adhere to or accumulate on the membrane surface or plug membrane pores by hydrophobic interactions, hydrogen bonding, van der Waals attractions, and electrostatic interactions [14,35,36,37,38,39]. As a result, the membrane separation process becomes more complex, and its permeability and selectivity are reduced. The latter leads to an increase in operating costs, energy demand, and shorter membrane lifetimes.

Thus, the current trend is to improve PPSU membrane materials and structures and to get membranes with both good separation and antifouling performance. Controlling the membrane surface properties and structure has been a common goal for improving membrane separation performance (Figure 1). Achieving this goal is not an easy task. However, various types of inorganic and organic materials have been used to improve the characteristics of the PPSU membranes [28,40,41,42,43]. Several features, including the PPSU and additive concentrations, molecular weights, the miscibility characteristics, the compatibility with organic and inorganic materials properties, and the solvent type, can impact the performance of these additives.

Multiple studies have reported the fabrication of PPSU-based membranes in different configurations, including flat sheet and hollow fibers [44,45,46,47,48]. However, to the best of our knowledge, there is no state-of-the-art report on PPSU-based membranes that summarizes their surface modifications and associated changes in performance. The current review serves to fill this research gap. More specifically, we bring together recent advancements in polymer and membrane development for the benefit of both academic and industrial researchers. We focus on various modification methods as well as performance evaluation. There are three main approaches for the modification of PPSU polymer or membranes with improved surface properties: (1) bulk modifications via sulfonation, amination, and chloromethylation; (2) blending with a synthetic polymer, inorganic nanomaterials, and biopolymer; and (3) surface modification via physical and chemical approaches (Figure 2).

## 2. Polyphenylsulfone Characteristics

PPSU is the abbreviation for Polyphenylsulfone. Also known as PPSF, PPSU is a new member of the sulfone polymers family that has multiple attractive properties such as high-temperature performance, good chemical resistance (maintaining its original properties after being exposed to a harsh chemical environment pH at 1–13), outstanding toughness, corrosion resistance, chlorine tolerance, excellent colorability, and very good dimensional stability [49]. The polymer can be distributed in two different families depending on the level of the molecular organization of the constitutive chains at the microscopic level. Compared to other sulfone polymers, PPSU is an amorphous polymer. Therefore, it features very good creep resistance, isotropic thermal and mechanical properties, and transparency. PPSU consists of an aromatic unit (phenylene) chain with a sulfone group and a benzene ring, connected by an oxygen atom. Due to this conjugated structure, the rigidity of the material can be maintained, and it gives good liquidity [28,50]. Figure 3 shows the molecular structures of PPSU.

The presence of the electronegative sulfone group results in sulfur being in its highest oxidation state. The latter brings excellent thermo-oxidative stability and easy functionalization. The surface of PPSU resin has a negative charge over a pH of 3 [30]. The existence of a biphenylene unit in PPSU resin significantly elevates the impact strength and reduces the notch sensitivity of the material. The latter results in strength at break (tensile) values greater than 75 MPa, a glass transition temperature of 288 °C, and a heat deflection temperature of 274 °C. Therefore, PPSU is expected to become the next widely used polymer in various applications, including membrane filtration, plumbing, food services, medical, aerospace, wire and cable, etc. On the other hand, the PPSU-based membranes that have been widely used in water applications have several drawbacks. The main drawback is related to its relatively hydrophobic nature. It reduces membranes’ permeability and makes them more susceptible to fouling during water treatment. Chemical cleaning is an essential step to sustain the membrane life. At the same time, high standards of water quality should also be met. Table 1 summarizes different methods that were used in the modification of PPSU membrane, modifier agents, and performance of modified membranes.

## 3. Bulk Modification

### 3.1. Polyphenylsulfone Sulfonation

Sulfonation is defined as an aromatic electrophilic substitution reaction used to attach the sulfonic acid group to the molecule of an organic compound via a chemical bond, wherein an ortho positions the aromatic ring in place of the hydrogen atom. It is attributed to the fact that this electron-donating oxygen atom activates the ortho position [35,40,44,105,106,107,108]. Sulfonation is the accumulation of sulfonic groups at the aromatic backbone (including a phenyl and a sulfone unit as part of the backbone) of PPSU due to electron-donating substituents enhancing sulfonation. However, electron repulsing substituents have the opposite effect. PPSU is difficult to sulfonate because of the electron-withdrawing effect of the sulfone linkages that deactivate the adjacent aromatic rings for electrophilic substitution [9,51,72,109,110,111]. Moreover, its sulfonation requires stronger reagents and/or longer times. However, sulfonated PPSU features increased hydrophilicity and proton conductivity in the presence of sulfonate groups in the polymer chain. The latter, however, can introduce negative charges.

The sulfonation of the aromatic backbone of the polymer is carried out before membrane fabrication using sulfonating agents. There are three groups of sulfonating agents that do not cause polymer chain degradation. Sulfuric acid (H_2_SO_4_), sulfur trioxide (SO_3_), chlorosulfonic acid (ClSO_3_H), fluorosulfonic acid (FHO_3_S), amidosulfonic acid (H_3_NO_3_S) and its complexes, halogen derivatives of sulfuric acid, etc., form the first group and are derived from sulfur trioxide. They are referred to as electrophilic reacting agents and are most frequently used to sulfonate aromatic compounds. The second group includes nucleophilic agents such as sulfites (SO_3_^2−^), hydrogen sulfites (HO_3_S^−^), and sulfur dioxide (SO_2_), which react with halogen derivatives and unsaturated compounds containing multiple bonds. The third group contains radically reacting agents. In particular, it includes sulfuryl chloride (SO_2_Cl_2_) and blends of gases (sulfur dioxide and chlorine (SO_2_ + Cl_2_), sulfur dioxide, and oxygen (SO_2_ + O_2_)). Sulfonation of polymer can be completed via either a heterogeneous reaction or a homogeneous reaction in hydrocarbons or chlorinated solvents.

The polymer sulfonation method works for most reagents. In the sulfonation protocol, the dried polymer is dissolved in a sulfonating agent and stirred at approximately 50 °C to produce a homogeneous solution in a nitrogen atmosphere. After the reaction, the solution is poured into a large volume of ice-cold deionized water under continuous stirring. As a result, a white precipitate is obtained. After standing overnight, the white precipitate is filtered and washed several times with cold deionized water to attain neutral pH level. The sulfonated polymer is then dried in a vacuum at room temperature [5,55,86,107].

A typical procedure is described by Hartmann–Thompson et al. [112] PPSU was added to dichloromethane (CH_2_Cl_2_), mixed, and placed in an ice bath on a stirring plate. Next, the solution was cooled to 10 °C under agitation. ClSO_3_H was added by drops over a one-hour period while stirring continuously. In the next step, acetic anhydride was added to the mixture by drops. The reaction was then allowed to continue for a period of time while stirring and maintaining the temperature. The reaction was stopped by gradually pouring the reacted solution into an ice-deionized water mixture. The resultant precipitate was recovered by pouring and washed repeatedly with deionized water until the wash water had a neutral pH. The PPSU was subsequently dried in an oven. Minor variations have been reported by other researchers, and a generalized overview of the sulfonation reaction scheme is shown in Figure 4a [107,113].

Liu et al. [9] demonstrated the sulfonation of PPSU random copolymers with various disulfonation levels. These were prepared via the direct polymerization method by aromatic nucleophilic substitution copolymerization. The reaction mechanism for this direct sulfonation has also been outlined. Karlsson et al. [52] prepared SPPSU by an anionic modification using n-butyllithium (BuLi) as a metalating agent. The preparation was performed via a one-pot synthesis in a reactor equipped with a gas inlet/outlet. In this method, the PPSU was dissolved in anhydrous tetrahydrofuran (THF) and cooled. The polymer solution was carefully titrated with a solution of BuLi until a faint reddish color was achieved. Subsequently, an amount of 2-sulfobenzoic acid cyclic anhydride corresponding to a twofold excess in relation to the lithiated sites of the polymer was quickly added in the form of a fine degassed powder and immediately dissolved and quenched the lithiated sites. Next, SPPSU was precipitated to remove the reactant residues via isopropanol. The precipitate was then filtered and dried in a vacuum and characterized by combining FTIR, ^1^H NMR, and ^13^C NMR spectroscopy. Licoccia and coworkers [55] followed the same methodology for sulfonated PPSU with H_2_SO_4_ and ClSO_3_Si(CH_3_)_3_.

H_2_SO_4_ is a low-cost sulfonating agent. However, it causes degradation of the main polymer chain when the reaction temperature is too high, or the reaction time is too long. This degradation may change the mechanical resistance of the membrane, therefore compromising its use in industrial applications. Among other reagents, the sulfonated PPSU with ClSO_3_Si(CH_3_)_3_ showed better sulfonation control when compared to the one sulfonated with SO_3_ [51,72]. However, SO_3_ has a drawback in that side reactions may occur. Moreover, the reaction is heterogeneous because when a part of the polymer reacts with SO_3,_ it becomes insoluble in an apolar solvent. The rest of the reaction must be carried out in a dispersed system and not in a homogeneous solution. To solve the heterogeneity problem, ClSO_3_Si(CH_3_)_3_ can be used.

### 3.2. Polyphenylsulfone Amination

In the core of sulfonic groups, the existence of an amine group can enhance the physicochemical properties of the membrane compared to the others formed by segments. This process involves a similar substitution reaction but with amine groups as substituents. PPSU can be simply nitrated to almost one nitrogen atom per reproductive unit via reactions with strong bases. The latter is due to the sulfonic group having a strong activation effect on the nitration process. In this nitration reaction, the acidic ortho-to-sulfone hydrogens are replaced with nitrogen atoms that result in a positive charge of the carbon atom in the phenylene unit. Arumugham et al. [84] used a two-step method for the amination of PPSU. The reaction scheme for amine-functionalized PPSU is shown in Figure 4b. In the first step, nitration of PPSU was performed with nitric and sulfuric acid. It resulted in nitrogen atoms being placed in the ortho position. In the second step, the intermediate was aminated using tin chloride with hydrochloric acid. A similar polymer amination procedure has also been applied by other researchers [114,115].

Aminated polymer synthesis was followed by the reaction in a nitrogen atmosphere. PPSU (10 g) was added to a mixture of sulfuric acid and nitric acid (1/4 mixing ratio, 200 mL) and then stirred for 2 h at 25 °C to produce nitrated PPSU (PPSU-NO_2_). The resulting product was washed with five times the deionized water (300 mL) and dried in a vacuum oven at 30 °C for 24 h. To synthesize aminated PPSU, the tin chloride (20 g, 0.105 mol) in hydrochloric acid solution (20 g, 37%) was added in 60 mL of ethanol in a circular bottom flask. The flask was kept at 70 °C and then allowed to stir for 10 min. Subsequently, the synthesized PPSU-NO_2_ (5 g) powder was added slowly to the flask. The color of the solution transformed from yellow to dark brown, indicating the progress of the reaction. The reaction was further carried out for 4 h with stirring at 70 °C. Afterward, the reaction mixture was poured into 400 mL of deionized water for precipitation. Finally, PPSU-NH2 was separated, washed with deionized water, and dried in a vacuum oven at 80 °C for 12 h [84]. Considered amino groups have a significant effect on the surface charge and hydrophilicity of the PPSU polymer, as was shown in ion exchange capacity and water absorption measurements.

### 3.3. Polyphenylsulfone Chloromethylation

PPSU is a polymer that has no functional groups for further chemical modifications. However, the chloromethylation reaction of aromatic polymers is of particular interest to researchers and includes attaching functional groups onto aromatic ring-like chloromethyl. Currently, chloromethylation is actively investigated, both theoretically and experimentally, in the context of the procurement of precursors for functional membranes. Chloromethyl generally provides higher flexibility with time since it can easily interact with any kind of amines. Once a functional group is attached to the aromatic ring, further reactions can occur, including immobilization of compounds for enhanced hemocompatibility resulting in antifouling capability [116]. Zhang et al. [64] carried out the chloromethylation of PPSU and cast anion exchange membranes from the resultant chloromethylated PPSU. Chloromethylation of PPSU was performed following the one-step procedure. In a typical reaction shown in Figure 4c, the PPSU was dissolved in tetrachloroethane, and then tin tetrachloride and chloromethyl ethyl ether were added to the solution. The reaction mixture was heated, and the temperature was maintained.

After the desired reaction time elapsed, the reaction mixture was precipitated in excess ethanol, and the chloromethylated PPSU polymers were isolated by filtration. The polymer was purified by dissolution in chloroform and precipitation with ethanol and then dried in a vacuum oven. Chloromethylation of the polymer was not easy to control, and the number of chloromethyl groups attached to the polymer could be very small. The latter affected the properties of the membrane. The reaction mixture often produced a gel if it was not properly controlled by adjusting the temperature and the reaction time. Most of the studies considered chloromethylation of the polymers with the system made from trioxane and chlorotrimethylsilane as an agent of chloromethylation in the presence of tin tetrachloride [117,118]. The chloromethylation of polymers was dependent primarily on the kind of chloromethylating agent, the polymer structure, the type and amount of solvent, the catalyst, and other parameters of the reaction.

## 4. Polymer Blending

Polymer blending is an emerging research area in polymer science and engineering that aims to improve polymer properties. Blending is the physical mixing of two or more polymers that belong to the same chemical family or to different ones, such as homopolymers or copolymers and organic-inorganic materials [119,120,121,122,123,124]. These materials have attracted researchers’ interest due to their ability to modify the properties of other materials for particular applications. Earlier researchers obtained polymer blends from natural materials [125,126,127]. This technique is straightforward, fast, and cost-saving compared to other tedious and time-consuming methods. Today, bottom-up and top-down approaches are among the most used. Polymer blending makes obtaining products with superior properties compared with pure polymer materials possible [128,129]. When blending with hydrophilic additives, the structure and chemical properties of PPSU membranes can be easily modified. Multiple modifications of PPSU blends were suggested in the last 15 years. The majority of these PPSU were blended with other polymers for the synthesis of new PPSU-based membranes. Figure 5 summarizes the number of articles on PPSU blends indexed using the Web of Science.

### 4.1. Polyphenylsulfone Blended with the Polymer

PPSU blends with other polymers are considered one of the most practical methods to modify the membranes. More specifically, they brought together the synergistic properties of different polymer materials into a new mixed matrix with targeted structural characteristics and performance. Such an approach allowed the modification to overcome the deficiencies of the individual PPSU polymers. Moreover, polymer-polymer blends are cost-efficient, simple to obtain, and reproducible. At the same time, PPSU blends have some limitations in mixing with other polymers. This drawback affects the miscibility of the blends at the molecular level [56,83,130,131,132,133,134,135]. Thus, three types of blending mechanisms can be identified. The first one involves the two polymeric materials with single-phase properties that are dissolved in each other at the molecular level. The process is attributed to physical forces and hydrogen bonding and is known as miscible blending [120,136,137,138]. The second blending mechanism assumes that the two polymeric materials do not dissolve in each other and form the interface between components. This mechanism is referred to as immiscible blending [20,53,139,140,141,142]. Finally, the third mechanism implies that one polymeric material partially dissolves into another polymeric material during the heterogeneous phase. This behavior is known as partially miscible blending [143,144,145,146]. Whereas a number of research reports exist on the blending of other polymers, these reports have described the concept of physically blending two or more polymers with PPSU to obtain the new PPSU product (Figure 6).

PPSU could be sulfonated using sulfonating agents such as sulfuric acid, sulfur trioxide, chlorosulfonic acid, and others. Attributable to the good miscibility, sPPSU could be blended with PPSU matrix or other polymers at any ratio. Arumugham and coworkers [49] developed modified PPSU UF membranes with higher fouling resistant properties. The membranes were treated sPPSU modifier. (Details are provided in Appendix A). The membrane casting solutions were prepared and contained various amounts of sPPSU to entangle with the PPSU matrix. UF membranes were prepared using the phase inversion technique. It was found that the modified PPSU membranes showed a very low tensile strength value because of the influence of different content of sPPSU on the polymeric chain stacking of PPSU.

However, the hydrophilicity of the modified membrane was significantly increased due to the existence of an H-bond interaction between a water molecule and the polar sulfonic acid group. The latter resulted in their superior surface properties, higher flux, and antifouling properties with better protein rejection. Hartmann–Thompson et al. [112] synthesized a polyhedral oligosilsesquioxane (POSS) nanofiller with various groups and then used it as a hydrophilic polymeric additive in the preparation of sPPSU membranes. When blending into sPPSU, the modified sPPSU composite membranes exhibited proton conductivity in combination with improved dimensional stability, heat resistance, and mechanical strength (Appendix A).

Similarly, Yong et al. [67] synthesized a high free volumes polymer with intrinsic microporosity (PIMs) by the condensation polymerization process. Then, they blended it with sPPSU to develop a dense membrane (Appendix A). This membrane can be used for gas separation. They found that the sPPSU blend with PIMs can significantly improve the antiplasticization properties and gas separation performance. The latter can be attributed to the sulfonic acid groups in the sPPSU polymer matrix having strong molecular interactions with CO_2_ and O_2_ and forming hydrogen bonds.

The PPSU membrane modification conditions were affected by the pore size and performance, such as PPSU/organic polymers membranes. Organic polymers accelerate the exchange rate between the solvent and the nonsolvent during the precipitation process. However, it strongly affects the size of the pores and the permeation rate. Organic polymers are often used as additives to blend with PPSU for improving membrane performance. Kiani et al. [66] modified the PPSU nanofibrous membrane by blending with polyethylene glycol (PEG) and prepared the membrane via the electrospinning process. The organic PEG polymers are soluble in water and may be eluded during application. A modified PPSU membrane showed the best behavior in all respects, including membrane porosity, hydrophilicity, and mechanical strength. Therefore, it was used for numerous practical applications. The PEG blending method also allowed improved antifouling properties. According to blending results, the addition of PEG to PPSU solution significantly increased the solution viscosity and caused a change in the shape of the nanofibrous beads from spherical to spindle-like. Similarly, the mechanical properties of the nanofibrous membranes were improved due to the blending of PEG. (Details are provided in Appendix A). The latter acted as a plasticizer, facilitated the mobility of PPSU polymeric chains, and reduced the brittleness of the modified membrane. In this approach, the modified membrane capable of reducing the fouling tendency by means of hydrophilicity improvement was considered [10,147,148,149].

Furthermore, the influence of strength of PEG/PPSU polymer interactions has been highlighted experimentally by Feng et al. [21]. They found that when blending with PEG organic additive, the PEG could interact with sPPSU in the form of hydrogen bonding. Moreover, it increased the viscosity of the sPPSU solutions due to enhanced polymer interaction and entanglement. In addition, with an appropriate amount of PEG blended with sPPSU solution, the modified membrane possessed improved mechanical strength, higher hydrophilicity, and permeation properties as shone in Appendix A. The authors observed that PEGs-rich sPPSU dope solutions experienced changes in dope viscosity caused by multiple factors. More specifically, PEG was a weak nonsolvent for the sPPSU molecules. Thus, when mixing with solvent, they decreased the solvent ability. The hydroxyl groups in PEG act as cross-linkers that connect sulfonic acid groups of the sPPSU polymer by hydrogen bonding. This blending process clarifies that the introduction of PEG into sPPSU blend solutions not only increases chain interaction and entanglement but also enhances the mechanical properties along with improving the membranes’ fouling resistance.

Another interesting study considered blending of the different polymeric pore-forming agents PEG and PVP, and a surfactant Tween-80 (Liu and Li [30]). They studied the asymmetric PPSU UF flat sheet membranes prepared by a nonsolvent-induced phase separation method with enhanced antifouling properties. The addition of the PEG, PVP, and surfactant to the PPSU solution significantly modified the morphology of the polymer. The cross-sections of PPSU membranes prepared by a single pore-forming agent had a tendency to form macrovoid structures near the bottom surface layer [78]. At the same time, the pore size distribution on the top surface of the polymer pore-forming agent (PEG and PVP) was not uniform due to the high mutual affinity of solvent to water and additives. The latter led to the formation of macrovoid structures. When blending the PPSU with polymeric pore-forming agents and surfactant, the modified membranes were observed (Figure 7). They featured uniform structures in the cross-section and top surface.

Moreover, the pores were more effective at the top surface of modified PPSU membranes, which became small and uniform. It was also found that the filtration resistance decreased, while the flux recovery ratio increased, and the cake layer and pore plugging resistance decreased. Yin et al. [13] investigated the effect of PVP molecular weight and concentration on the blending with PPSU solution and the performance of modified PPSU membrane, including morphology, mechanical strength, tensile strength, molecular weight cutoff (MWCO), and permeation. The PPSU–PVP blend membrane showed good alkali resistance and higher water flux compared to the pure PPSU membrane. The latter was attributed to stronger antialkali and hydrophilic properties. The PVP with high molecular weight and concentrations had a greater effect on membrane morphology compared to the counterpart with lower molecular weight. The latter was due to the increase in blending solution viscosity and the polymer becoming thermodynamically stable. It resulted in delayed demixing followed by membrane structure shift from fingerlike to spongelike (Figure 8). Furthermore, it can be pointed out that the mechanical strength increased with appropriate PVP molecular weight (360 kDa) due to suppressed formation of macrovoids. However, when the molecular weight of PVP was much higher (1300 kDa), the mechanical strength was significantly weakened. Gronwald et al. [96] synthesized a random poly (alkylene oxide) based on tri and multiblock copolymer additives by reacting hydrophobic PPSU blocks in a conventional process and then blending it with PPSU to fabricate hydrophilic flat sheet and single bore UF membrane for antifouling.

Polyimide is a representative of high-temperature engineering polymers in which the imide group is an important part of the molecule. The amide group is formed by a condensation reaction of an aromatic anhydride group with an aromatic amine. This group is responsible for forming incredibly strong and astoundingly heat and chemically resistant polymers [130]. Due to the superior properties of a polyimide polymer, it was often blended with different materials into binary blends and used for membrane fabrication. Jansen et al. [62] fabricated the PPSU/polyamide blend membranes with wet phase inversion induced by an immersion-precipitation technique for solvent-resistant nanofiltration applications. The organic solvents flux through the prepared membranes was increased by blending PPSU with polyimide. Moreover, the gas separation properties of the PPSU/polyamide blend membranes were improved.

### 4.2. Polyphenylsulfone Blended with the Nanomaterials

Apart from introducing blends with polymers and copolymers, inorganic nanomaterials are another promising modifier of PPSU. The blending of inorganic materials with the PPSU matrix has become an attractive methodology for the improvement of polymeric membranes and thus has attracted researchers’ attention in the recent decade [41,97,150,151]. Multiple studies were carried out on the development of nanocomposite PPSU membranes by the combination of inorganic nanoparticles. The incorporation of inorganic nanoparticles could produce a barrier that prevented free radicals from attacking the PPSU backbone. For example, the blended inorganic nanoparticles in the PPSU matrix have been reported to improve the polymer properties, mostly by: (a) enhancing the mechanical and thermal properties; (b) inducing some new functional properties into the membrane; (c) improving structural morphology; (d) enhancing the mass transfer; (e) increasing the larger effective membrane surface area and follow-on greater permeability; (f) improving the surface charge; (g) raising a membrane’s hydrophilicity as well as antifouling properties. Up to the present time, various types of inorganic materials have been blended as additives in the PPSU matrix. These included graphene oxide, copper oxide, titanium dioxide, zinc oxide, magnesium oxide, zirconium oxide, silicon dioxide, activated carbon, carbon nanotubes, and metal-organic frameworks. As reported, the properly blended nanoparticles can facilitate the dispersion in a PPSU polymer solution and cause a possible rearrangement of nanoparticles in the membrane matrix (Figure 9).

Shukla et al. [152] made blends of PPSU with graphene oxide to fabricate a nanocomposite UF membrane with increased hydrophilicity. They showed that this method was effective because only a small weight percentage range of the nanoparticles was required to modify the PPSU membrane properties while maintaining the structural properties unaltered. As a result of less agglomeration, excellent nanoparticle dispersion in the membrane was achieved with optimized concentrations. The dispersity problem that occurred during the fabrication of nanocomposite membranes was solved by optimizing nanoparticle concentrations. Significantly higher water permeabilities were observed, along with higher negative surface charge properties. The latter was explained by antifouling properties, thereby improving the fouling resistance ability up to 58% ± 3% (Appendix A).

Sani and coworkers [65] self-synthesized copper-1,3,5-benzenetricarboxylate (Cu-BTC) nanoparticles with different contents to blend with PPSU for improving methanol separation ability and nanofiltration performance. The Cu-BTC particles were enriched at the blend membrane surface. By incorporating Cu-BTC particles, the performance of the blend membranes was significantly improved (Figure 10).

The membranes showed that the incorporation of Cu-BTC with low loads tended to have a smaller molecular weight cutoff than that of PPSU. However, increasing the content of nanoparticles with PPSU led to a smaller surface pore size but better separation efficiency. The enhancement in membrane flux and dye–methanol separation at lower Cu-BTC loadings could be ascribed to the good dispersion of the nanoparticles in the PPSU blends coupled with their improved interfacial contact with the polymer matrix. Then, they also fabricated organic solvent (methanol, ethanol, isopropanol, acetonitrile, ethyl acetate, n-hexane, and n-heptane) -resistant nanofiltration membranes using Cu-BTC with a PPSU blend. The PPSU nanocomposite membrane exhibited acceptable durability and dye/methanol solution separation performance stability [70,153].

Functionalized activated carbon (FAC) and multiwall carbon nanotubes (MWCNTs) could be directly blended with PPSU to prepare membranes for improving the membrane separation. Saranya et al. [68] fabricated composite membranes from the blends of PPSU and FAC by the wet-phase-inversion method. The structural morphology of PPSU/FAC membranes clearly showed a composite structure with a porous sublayer. The results demonstrated that chemical modifications of nanoparticles resulted in better dispersion of nanoparticles in the membrane with reduced agglomeration. The dispersity problem that occurred during membrane fabrication was solved by modifying nanoparticles. Next, the hydrophilic PPSU/FAC composite membranes were evaluated for the adsorption of phenol and revealed that the smaller the blending of FAC in PPSU, the higher the adsorption of phenol. Thus, the better dispersion of minimal blending offers higher accessibility to adsorptive sites (Appendix A).

Nayak et al. [94] prepared asymmetric mixed matrix ultrafiltration membranes from the blends of PPSU and MWCNTs to inspect the heavy metals separation efficiency from the aqueous media. By blending the MWCNTs in the PPSU matrix, the morphology of the fabricated membrane showed heterogeneous layers that consisted of a dense skin layer on top and a porous supportive sublayer and allowed significant improvement of surface roughness. Apart from the surface properties, the membranes featured better antifouling ability and exhibited good rejection performance for heavy metals. Shukla et al. [87] also modified the PPSU membrane by blending Ag-MWCNTs and the modified nanocomposite membranes. They showed good ion removal and antibacterial capacity due to the functional groups of additives. The Ag-MWCNT/PPSU nanocomposite membrane has been reported for potable water purification applications due to its excellent properties and relatively good compatibility. (Details are provided in Appendix A).

A nanomaterial was blended with PPSU to prepare nanocomposite membranes. The nanocomposite membrane reduced biofouling abilities and better water fluxes. The possibility of bacterial attachment (biofouling) was associated with water transport and surface characteristics [87,95,104]. The casting solution composition, concentration, and inorganic salts all had an impact on the membrane filtration efficiency. The modified membrane was compared to a nanomaterial and the surface charge membrane with respect to their potential to inhibit bacterial growth and biofouling of a nanocomposite membrane. Shukla and coworkers [41] modified the PPSU UF membranes by the incorporation of carboxyl-functionalized graphene oxide using the phage inversion technique. The membrane characteristics for biofouling were tested in the presence of Gram-negative bacteria (*E. coli* and *P. aeruginosa*) and Gram-positive bacteria (*S. aureus*). It was found that the nanocomposite membrane inhibited the attachment, colonization, and biofilm formation of bacterial species. As a result, these modified membranes may be more resistant to biofouling.

A wide range of metal-oxide-based nanoparticles have a large surface area and specific functional groups and thus could be blended with PPSU to modify the PPSU membrane. The nanoparticles exhibit enhanced properties at a nanoscale level, and the PPSU matrix is used to hold the nanoparticles together. Blending nanoparticles with the PPSU polymer allowed for improved membrane properties, including morphology, thermal and mechanical properties, corrosion rate, oxidation resistance, and surface functional groups. D’ıez-Pascual and coworkers [63] and Dass et al. [77] investigated the effects of titanium dioxide (TiO_2_) nanoparticles blended with PPSU and sPPSU to obtain antibacterial and antifouling membranes and found that hydrophilic groups were mostly concentrated at the membrane surface. The modified membrane revealed the existence of strong hydrogen bonding interactions between the sulfone group of PPSU and the hydroxyl moieties of the nanoparticles, which were homogenously dispersed within the polymer matrix without adding coupling agents. The attachment of nanoparticles and polymer matrix tailored the surface chemistry of the composite membrane by altering the morphology and water permeability. The approach of Kumar et al. [45,103] was used to fabricate hollow fiber membranes for the removal of arsenic from aqueous media. The authors have blended a binary zinc-magnesium oxide (ZnO-MgO) on cellulose acetate (CA) with PPSU as well as zirconium oxide (ZrO_2_) with PPSU. The latter allowed evaluation of the affinity of the PPSU to the nanofiller and revealed significant enhancement in the overall performance of the membrane with water permeability and rejection. Arumugham et al. [69] selected a MgO/sPPSU/PPSU blend and a sequence of addition where the magnesium oxide (MgO) nanoparticles were first mixed with sPPSU, and this composite was then mixed with PPSU. As a result, the castor oil/water emulsion separation and antifouling properties were improved. Another interesting blending method to quantify the PPSU interactions and relate them to the distribution of the silica nanoparticles was suggested by Dehban et al. [97]. They studied blends of a PPSU matrix with SiO_2_ and then fabricated the nanocomposite ultrafiltration membranes by a combined vapor induced phase separation and nonsolvent induced phase separation technique. The polymer concentration allowed the researchers to determine which range of nanoparticle concentration showed better performance. It showed the effect of SiO_2_ nanoparticle concentration on the low and high PPSU matrix concentrations in blend solution. While the PPSU concentration in the blend solution was low, the increase of SiO_2_ nanoparticle concentration brought about the lower fluxes. However, at high PPSU concentrations, the increase in the nanoparticle concentration resulted in an increase in permeate flux. Similar observations were made by Isloor et al. [89], who blended PPSU with nano tin oxide (SnO_2_) and found an increase of hydrophilicity with SnO_2_ concentration. The rejection rate of mixed matrix hollow fiber nanocomposite membranes has improved.

Metal-organic frameworks (MOFs) blended with PPSU polymer allow enhanced general membrane performance, and thus various blending techniques for the preparation of PPSU composite membranes have been actively developed. In contrast to blending with the PPSU matrix, Xiao et al. [100] used as-synthesized MOF-CAU-1 nanoparticles as a filler and PPSU as a polymer matrix. The PPSU solution blending with different amounts of MOF-CAU-1 allowed fabricating a mixed-matrix membrane using immersion-precipitation and phase transformation techniques. The authors argued that the blending with MOFs resulted in a typical asymmetric membrane structure and higher surface roughness with nodular structure appearing on the surface. The abovementioned modified PPSU results illustrate that blending with MOFs can play a dominant role in pure water flux and antifouling performance. Recently, Shukla et al. [102] found that introducing zinc-based MOFs filler in PPSU blends led either to the increase or to the decrease in the performance of the membrane depending on the filler concentration. The presence of filler was shown to result in different compositions of the membrane due to the stronger interactions of the PPSU matrix with the zinc-based MOFs. The authors found that the surface charge and pure water fluxes of the membranes were much greater in the presence of filler (Appendix A). Moreover, the water permeation and protein rejection were improved. Finally, the antifouling analysis confirmed that the membranes exhibited fewer tendencies for fouling.

### 4.3. Polyphenylsulfone Blended with the Biopolymer

Apart from introducing blends with polymers and copolymers, a number of interventions have been used to improve the polymeric membrane properties during their applications. One methodology implied creating such membrane modification via blending PPSU polymer with hydrophilic biopolymers. The benefit of blending with hydrophilic biopolymers is that modification takes place even within the PPSU membrane pores and not only on the surface. The naturally occurring biopolymer containing the different functional groups can be used to modify the PPSU hydrophilicity and surface structures. There is growing interest in polymer technology to use biopolymers as modifiers or as additives due to their biocompatibility, nontoxicity, and antimicrobial properties [127,154,155,156,157,158]. In recent studies by Alam et al. [74] the PPSU polymer blended with naturally occurring carrageenan biopolymer as an additive facilitated improvement in membrane porosity. The authors observed that the water-soluble carrageenan was unavoidable at the membrane surface after modification. It can be pointed out that carrageenan is a very promising modifier of PPSU polymer. In particular, it allows us to study the membrane properties and characterize it with contact angle, zeta-potential, porosity, mean pore diameter, and water permeability.

## 5. Polyphenylsulfone Surface Modification

The top selective layer plays a significant role in PPSU membrane performance. The membrane surface (top layer) properties can be correlated with the hydrophilicity/hydrophobicity, surface charge, and surface roughness/smoothness. These characteristics define the affinity of the membrane top selective layer toward the applications. Hence, numerous research efforts have been devoted to the modification of membrane surface properties using physical or chemical modification processes [159,160,161,162]. In a physical surface modification, the modifiers interacted with a top layer of the polymeric membrane surface and were attached by van der Waals attraction, hydrogen bonding, or electrostatic interaction [163,164,165,166]. This approach allowed long-term operation. In a chemical surface modification, the modifiers are connected to the polymeric membrane surface through covalent bonding. In this process, the polymeric chain was activated by chemical reaction or high-energy radiation and followed by the addition of the modifier. Thus, the membrane bulk was not significantly affected. However, the membrane surface properties were significantly improved and demonstrated better chemical and structural stabilities [21,83,167,168,169]. We further review the techniques of physical and chemical surface modification of the PPSU membrane surface in Figure 11.

### 5.1. Physical Modification

Physical surface modification of PPSU membranes were investigated by surface coating or surface adsorption. Surface coating is a convenient and inexpensive method to modify the membrane surface. It has been generally used in membrane industries for large-scale production [147,159,170]. This method involves two general steps. First, the coating material is sprayed as a liquid or laid down as a surface of the porous membrane, which is then covered and bonded to the surface. Second, we evaporate the residual solvent/water at a moderate temperature. The surface-coated layer can be controlled by adjusting the operating parameters. Surface coating is done by a relatively simple process that creates a functional surface layer on the membrane surface. Surface coating is widely used elsewhere. For instance, Çalhan et al. [36] coated the polydimethylsiloxane (PDMS) with PVP additive and with two fumed silica fillers onto a PPSU support layer membrane surface, followed by cross-linking PDMS to improve their separation property. The membrane surface hydrophobicity increased with the spreading of PDMA chains over the PPSU membrane surface. Moreover, the addition of silica fillers with PDMA to the active layer allowed a further increase of hydrophobicity. Additionally, an increase in the PDMS thickness significantly affected the flux and the separation rate. For the separation performance, the butanol fluxes increased with increasing butanol concentrations in feed, while the separation factor showed an opposite trend for PDMS with additives. The membranes with PDMS and silica particles embedded into the support layer slightly increased the separation. At the same time, fluxes were not affected compared to the pure PDMS-coated membrane.

Tashvigh et al. [79] investigated the effect of coating of ionically cross-linked hyperbranched polyethylenimine (HPEI) on the sPPSU membranes. This surface modification method included several stages. First, the dope solutions were prepared by dissolving sPPSU and HPEI in DMF/THF. Next, they were used to cast films on a glass plate and develop the sPPSU/HPEI membrane. Afterward, the HPEI in water or ethanol solution was coated on the prepared sPPSU/HPEI membrane to enhance the rate of the ionic cross-linking reaction and seal the membrane defects under different conditions. Appendix A shows the ionic cross-linking reaction between sPPSU and HPEI polymers.

The authors applied the dip coating and pressure-assisted coating methods. In the dip coating, the membrane was immersed in HPEI-water or ethanol solution for 30 min and washed with water to remove any excess HPEI. For the pressure-assisted coating approach, the membrane was placed in a dead-end permeation cell, and the HPEI-ethanol solution was filtrated for 30 min under 5 bar. Next, pure water filtration for 30 min was used to remove the excess HPEI and ethanol from the surface. The surface coating procedures used dip- and pressure-assisted coating. The surface-coated membranes demonstrated good chemical stability in ethanol and isopropanol. In addition, coated membranes were more stable in terms of pure ethanol permeability and rejection of different dyes.

Recently, Alam and coworkers [104] performed another physical coating study on the surface of the PPSU substrate membrane with polyaniline (PANI) and consisting of a thin-film nanofiltration membrane. In their method, the liquid PANI mixture (ammonium persulfate served as the oxidant, and HCl was used as a doping agent) was first coated on the surface of the PPSU substrate membrane using the pouring method and then kept for the polymerization process (Figure 12). The PANI coating significantly reduced the membrane surface roughness with a substantial change in the lower contact angle. Moreover, the polyaniline thin-film coated PPSU membrane had a less negative surface charge compared with the PPSU substrate membrane. However, the backbone of the PANI corresponding to the hydrophilic region extended to a water phase and repelled various foulants in feed water. Experiments using a dye separation and antimicrobial activity (such as *Escherichia coli* (*E. coli*) and *Staphylococcus aureus* (*S*. *aureus*) showed that the PANI-surface modified membrane improved organic methylene blue dye rejection and antimicrobial capability.

### 5.2. Chemical Modification

PPSU membrane surface modification has been considered as a suitable process to improve the surface properties of the membranes without affecting bulk properties. The chemical surface modification includes multiple processes such as plasma treatments, thermal-induced lamination, UV-induced grafting, interfacial polymerization, and others. These processes are commonly used due to their relative simplicity, high density, and wide-ranging monomers. Moreover, these monomers can be polymerized via covalent bonding interactions between the polymer chains with new functionality and the membrane surface (Figure 13). The latter can produce membrane surfaces with desirable physicochemical properties.

The plasma treatment has been widely used to achieve hydrophilic surfaces throughout the membrane structure. Plasma treatment has several advantages, including waste-free processes, fast reaction time, and high flexibility [171,172,173,174,175]. The plasma treatment process only requires the use of a lower degree of ionization and is usually referred to as a cold plasma process [176,177]. It introduces the different functional groups on the membrane surface with the variation of plasma treatment parameters such as chemical properties, power, and flow rate of the plasma gas and precursors along with treatment duration. Wang et al. [91] modified an electrospun PPSU nanofiber membrane using heat and plasma treatments. Thermal treatment results revealed that the mechanical properties of the modified membranes were affected, depending on the treatment. They showed that the thermal treatment transformed a loose nonwoven nanofiber structure into a robust interconnected PPSU network. Thus, an increase in both stress and strain was achieved. Moreover, the plasma treatment can be used to improve the wettability of the membrane surface. The low-pressure plasma-treated membrane surface turned superhydrophilic. In another work, Norrman and coworkers [11] carried out the modification of electrospun PPSU nanofibrous membranes via two different types of plasma. First, they applied low-pressure microwave plasma and then used atmospheric-pressure coplanar barrier discharge to control the surface chemistry and optimize hydrophilicity. The composition of chemical anchor groups for plasma-treated PPSU was monitored by X-ray photoelectron spectroscopy. They showed that the atmospheric-pressure plasma treatment provided subtle oxidation. Moreover, the low-pressure plasma provided significant oxidation that resulted in PPSU nanofibrous surfaces with very high hydrophilicity.

Thermal-induced lamination is a simple way to modify the PPSU membrane surface. Recently, Kiani et al. [147] modified a PPSU porous support surface via a thermally-induced lamination. In this approach, the nanofibrous support was immersed in the water bath containing the floating PPSU thin film. Subsequently, moving the support upward and toward the thin film, it was taken out of the water while the thin film was placed on the surface of nanofibrous support. The acquired membranes were further heated to induce the adhesion of fibers and the addition of the thin film and the nanofibrous support. Heat treatment was carried out in an oven at 245 °C for one hour. The efficiency of the modified PPSU membranes was increased in pure water flux and steady permeate flux without sacrificing the rejection rate.

One more approach to modify the PPSU membranes is UV-induced grafting. It is attractive due to its simplicity and low cost. Additional advantages include the possibility of being applied to existing membranes and further functionalizing through postreaction [178]. Moreover, a wide range of monomers can be grafted. UV-induced grafting requires a low temperature and mild reaction conditions. Zhong et al. [58] applied the UV-induced grafting method for the first time to modify sPPSU membranes. It was highlighted that the effects of UV-initiated reactions onto properties of sPPSU membranes would mostly depend on the membrane structure and surface properties. The authors used the two different types of positively charged grafting monomers: (2-(methacryloyloxy) ethyl) trimethyl ammonium chloride and diallyldimethylammonium chloride. It is worth noting that after the UV-induced grafting, the morphology of the membrane top surface and selective layer thickness changed significantly. The slow change from a dense to the porous substructure with no signs of delamination demonstrated strong covalent chemical bonding between the sPPSU of the substrate and the vinyl monomers. In addition, the longer the UV exposure was, the thinner the membrane selective layer formed. The latter was due to the existence of an additional unsaturated double bond on diallyldimethylammonium chloride monomer that resulted in significantly improved separation performance.

The interfacial polymerization process is used to modify a PPSU membrane surface because of its low cost, availability, resistance to compaction, relatively high hydrophilicity, and chemical tolerance of the surface to a wide range of pH [100,179,180]. The surface modification process involves two monomers and includes several steps. First, a porous PPSU support is soaked in an aqueous solution of an m-phenylene diamine (MPD). Second, the amine-impregnated PPSU membrane is immersed in a solution of trimethyl chloride (TMC) in hexane. Finally, the PPSU membrane is cross-linked by performing heat treatment. The membrane modification via interfacial polymerization process is shown in Figure 14. Based on this surface modification process, the various factors affecting modified membrane surface properties were identified. These included structural morphology, posttreatment conditions, solvent type, the concentration of monomers, reaction time, and reaction curing temperature.

Liu and coworkers [59] investigated the separation performance for sulfonated thin-film composite NF membrane fabricated via interfacial polymerization. First, the surface of the PPSU membrane was pretreated with oxygen plasma to increase adhesion properties regardless of the material being used. Moreover, it resulted in the enhancement of the physical adsorption of sulfonated aromatic diamine monomers. Next, interfacial polymerization was performed. It was done at the microporous PPSU support membrane using two types of sulfonated aromatic diamine monomer, namely, 2,5-bis (4-amino-2-trifluoromethyl-phenoxy) benzene sulfonic acid and 4,4-bis (4-amino-2-trifluoromethylphenoxy) biphenyl-4,4-disulfonic acid. Moreover, piperazine (PIP) with TMC was also used. The thin film formed at the PPSU support under the optimum condition showed an increase in water flux. In particular, surface hydrophilicity was enhanced by the presence of sulfonated aromatic diamine monomers without compromising the rejection. Widjojo et al. [61] modified the surface of the thin-film composite of a sPPSU supporting membrane via interfacial polymerization process using an MPD and TMC. The authors managed to enhance the performance of forward osmosis membrane applications. The thin layer morphology of the membrane PA surface had a spongelike structure without the formation of macrovoids. The latter provided better mechanical stability for the sPPSU membrane in the long-term perspective. The polyamide (PA) skin layer of the modified sPPSU membrane significantly improved the hydrophilicity and allowed it to achieve water flux under pressure-retarded osmosis mode. Golpour et al. [81] modified the PPSU-graphene oxide (GO) support layer to make a thin-film nanocomposite membrane by interfacial polymerization with PIP and TMC monomers. It was observed that embedment of nanoparticles (MOF) into a PIP monomer during interfacial polymerization significantly changed the membrane surface chemistry and morphology, leading to an improvement of hydrophilicity, surface charge, thermal stability, and mechanical strength. It should also be noted that the surface modification with nanoparticles affected membrane pore size and pore size distribution [76]. Moreover, the surface-modified membrane exhibited remarkable improvement in the antifouling capability and excellent long-term stability.

Recently, Shukla et al. [181] have studied the desalination performance and chlorine resistance of thin-film nanocomposite membranes incorporating Zn-MOFs synthesized by interfacial polymerization over PPSU support membranes with amines such as trimesoyl chloride and phenylenediamine. The nanocomposite polyamide layer formed over the PPSU supports showed enhanced salt-rejecting properties with a comparatively lower salt permeation rate. Moreover, the membranes demonstrated significant water stability during filtration and chlorine resistance after a chlorine-soaking test due to the superior compatibility between the polyamide and Zn-MOFs on the PPSU supports.

## 6. Conclusions and Future Prospects

This review article has summarized up-to-date methods used for the modification of PPSU membranes. The modification methods vary with the materials used and the processes applied. These factors influence membrane stability and functionality. There are three main approaches for the modification of PPSU membranes: (1) bulk modification; (2) blending; and (3) surface modification. Based on these methods, the modified PPSU membranes with unique properties can be directly obtained.

As stated above, bulk modification and blending are among the most used methods. Bulk modification can include amination and chloromethylation. However, the most frequently used method is sulfonation. The sulfonation can be carried out through sulfonating modifier agents by the accumulation of sulfonic groups at the aromatic backbone of PPSU without polymer chain degradation. Moreover, it is very important to select reactive modifier agents that have high compatibility with the PPSU polymer matrix to avoid negative effects on the properties of the membrane. Doing so allows the sulfonation of bulky groups in the polymer chain structure, placement of such groups, and provides more flexibility to the polymer backbone. Blending is another promising approach to PPSU modification. It is relatively simple and has been often used with inexpensive materials to obtain superior characteristics. However, it has limited applicability due to immiscibility with hydrophilic and hydrophobic materials. The commercially available and synthesized biopolymers, organic and inorganic, have been widely blended directly with the PPSU matrix. As a result, the developed membranes possessed specific intrinsic chemical, mechanical, and physical properties. Polymeric additives such as carrageenan, PEG, and PVP are well known as pore formers. Moreover, they can enhance the characteristic asymmetric structure of membranes developed in the phase inversion process. With the same motivation, the organic and inorganic blending materials such as PEI, SPEEK, MOFs, ZnO, TiO_2_, and CNTs, and others have been proposed to enhance surface charge, improve hydrophilicity, modify pore structure, and add antibacterial capability to metal-based inorganic materials. Among the newly arisen inorganic additives, PPSU blends with GO are of superior functionality and permeability and can overperform the traditional PPSU polymers.

The performance of existing PPSU membranes can be improved through surface modifications, including physical or chemical modification processes. Surface modification may take place by bringing specific moieties via the surface-coating, plasma treatment, thermal-induced lamination, UV-induced grafting, interfacial polymerization, and other methods. Many of these surface modification methods have limited applicability due to high cost, complicated operation procedures, or difficulty in scaling up. In contrast, the interfacial polymerization method is widely used in commercial membrane production. Compared with surface-coating, plasma treatments, and UV-induced grafting, interfacial polymerization is a more promising approach. In particular, it allows achieving covalent linkage between membrane surface and modifiers. Membrane surface modification, either by physical or chemical processes, generally leads to additional resistance due to active layer formation after modification. The latter slows water permeation through the membrane active layer and decreases water permeability. Hence, the surface-modified membrane active layer should be thin, and the trade-off of flux reduction and surface properties should be optimized and balanced with modification conditions.

In the last decade, tremendous progress has been made in PPSU membrane modification to improve its surface properties. However, there are some challenges remaining for the future industrial applications of PPSU membranes. It is worth noting that developing ‘ideal’ methods for surface modification of membranes is a relatively challenging task. It is attributed to the inevitable compromise between exceptional physical and chemical properties and the possibility of long-term commercial applications. An in-depth understanding of the surface structure-property relationship between the membrane surface and modifiers during cross-linking is essential. Not only flat-sheet membranes, but also hollow fiber membranes, can be modified with this method. In part, this problem can be tackled using transport models and numerical simulations. Moreover, the polymer blend can be further improved to combine with other modifiers such as biomolecules and inorganic nanofillers through proper compositions. Innovations will influence the development of better membrane systems that eliminate the trade-off effect and are suited for long-term and commercial applications.

## Figures and Tables

**Figure 1 membranes-12-00247-f001:**
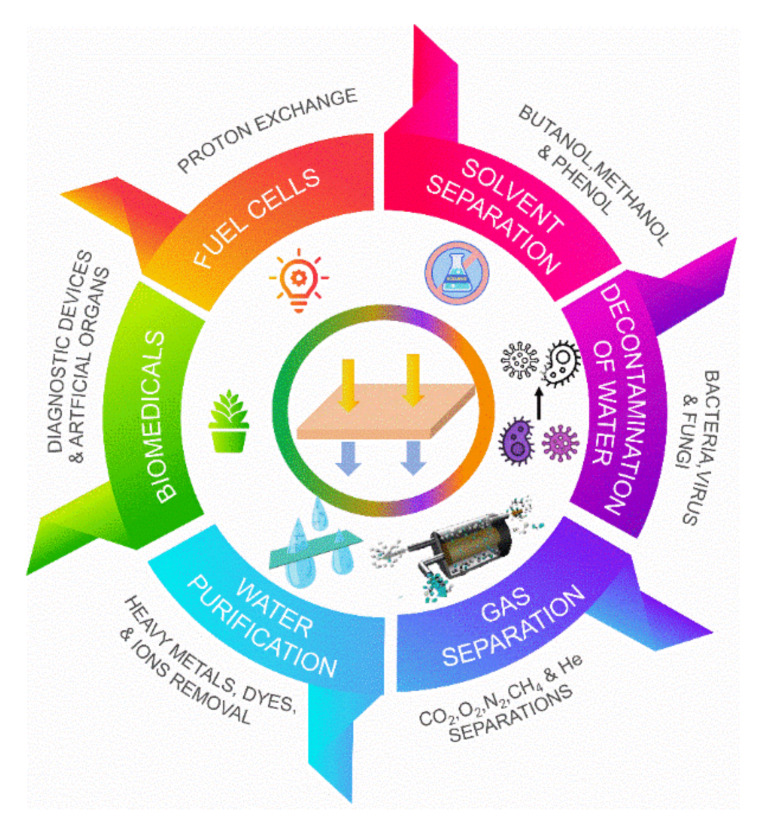
PPSU membranes significant environmental applications: water purification, gas separation, decontamination of water, solvent separation, fuel cell, and biomedical.

**Figure 2 membranes-12-00247-f002:**
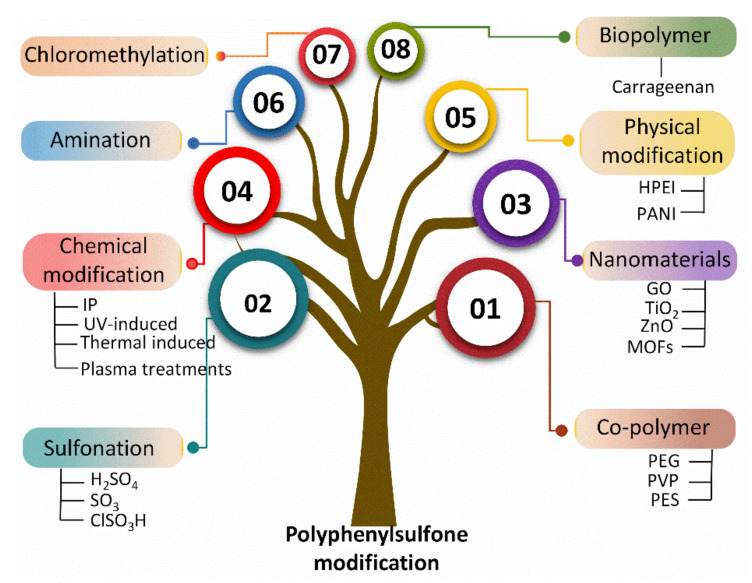
Modification techniques for PPSU utilizing various modifiers.

**Figure 3 membranes-12-00247-f003:**
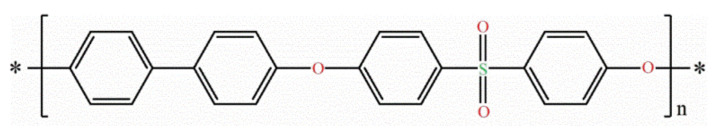
General structure of Polyphenylsulfone polymer.

**Figure 4 membranes-12-00247-f004:**
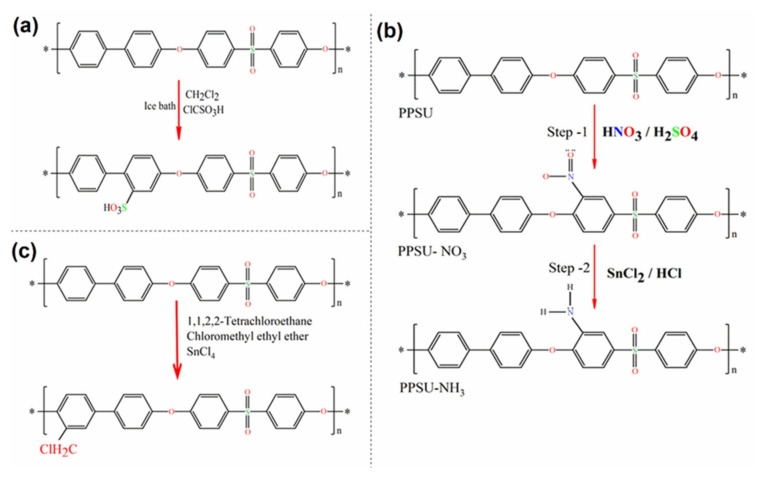
Synthesis of polyphenylsulfone. Reaction pathways: (**a**) sulfonation; (**b**) amination; (**c**) chloromethylation.

**Figure 5 membranes-12-00247-f005:**
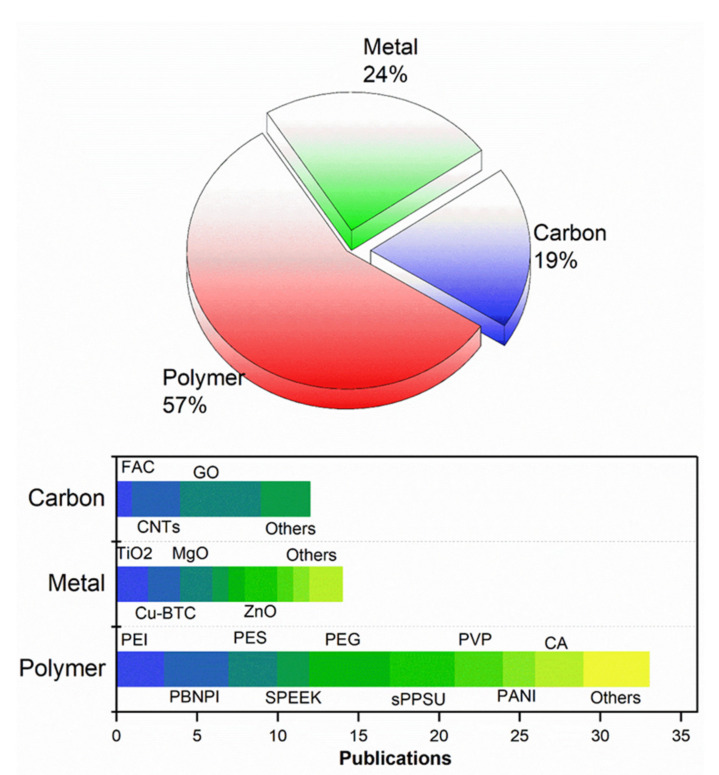
A number of articles on polyphenylsulfone blending indexed in Web of Science for “PPSU blend membrane”. Search results and details are provided in Table 1.

**Figure 6 membranes-12-00247-f006:**
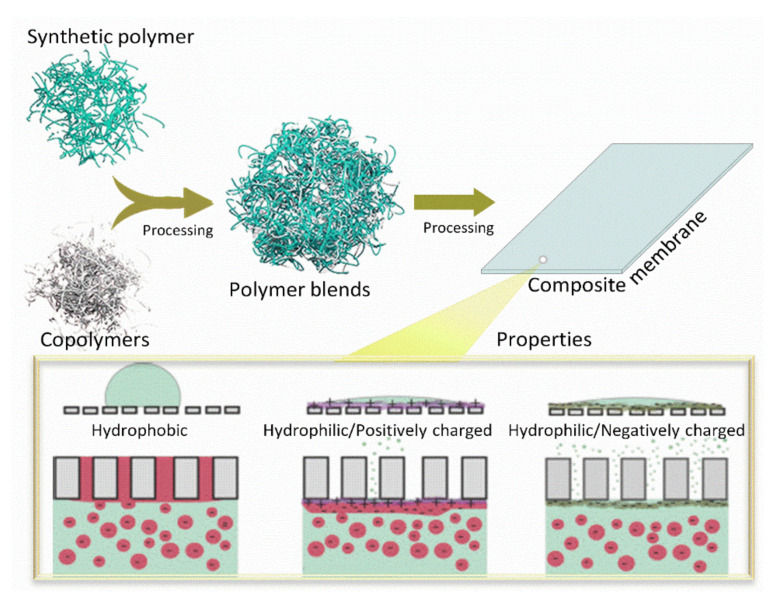
Schematic illustration of the preparation of PPSU membrane with superior properties via a typical polymer and copolymer blend system.

**Figure 7 membranes-12-00247-f007:**
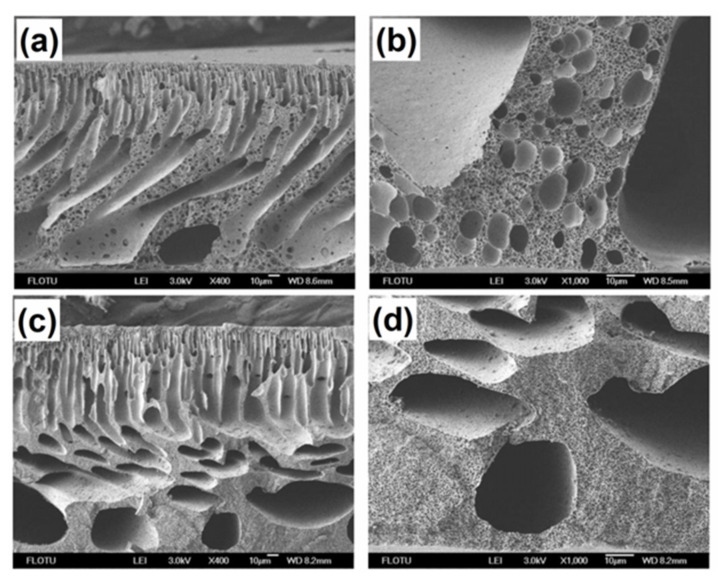
SEM photomicrographs of cross-sections of PPSU UF membranes prepared with: (**a**,**b**) PVP-15000; (**c**,**d**) PEG-6000 as a polymeric pore-forming additives.

**Figure 8 membranes-12-00247-f008:**
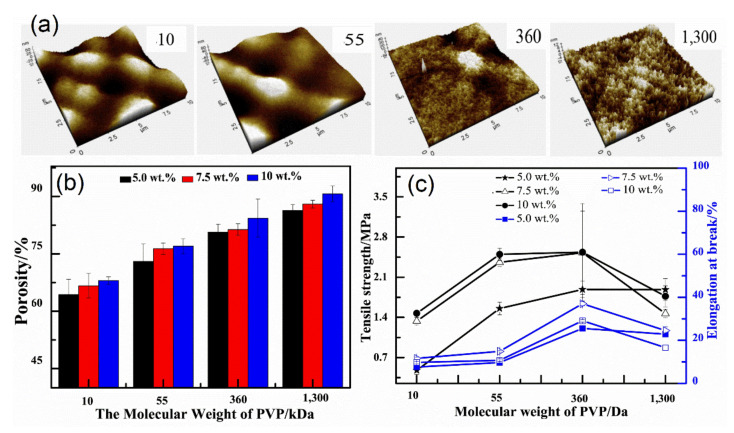
PPSU UF membranes with different molecular weight of PVP: (**a**) AFM images; (**b**) porosity; (**c**) tensile strength.

**Figure 9 membranes-12-00247-f009:**
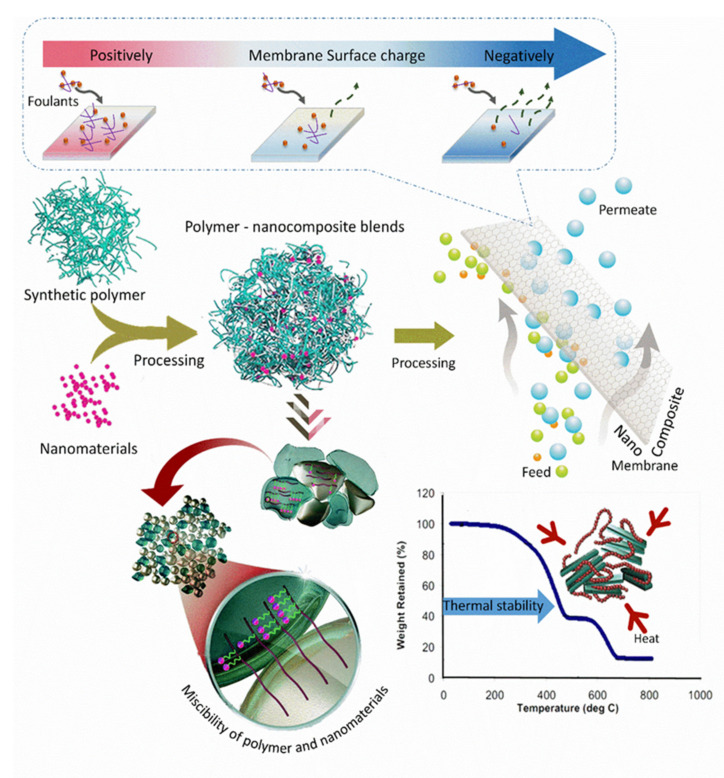
Illustration of the preparation of a PPSU nanocomposite membrane with higher surface and thermomechanical properties via a typical polymer and nanomaterial blend system.

**Figure 10 membranes-12-00247-f010:**
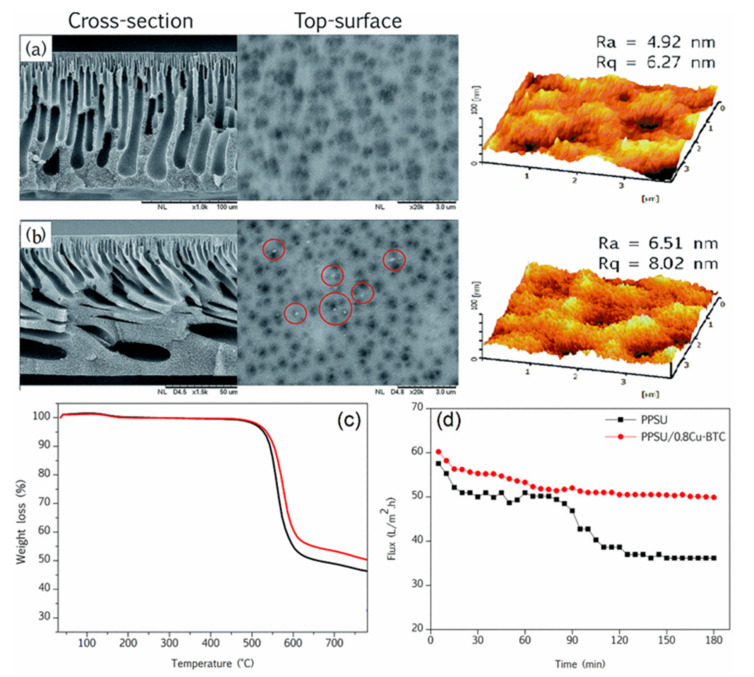
PPSU membranes embedded with Cu-BTC; SEM images of cross-section, top surface and 3D AFM images: (**a**) PPSU; (**b**) PPSU/Cu-BTC; (**c**) TGA curves; (**d**) flux profile membranes.

**Figure 11 membranes-12-00247-f011:**
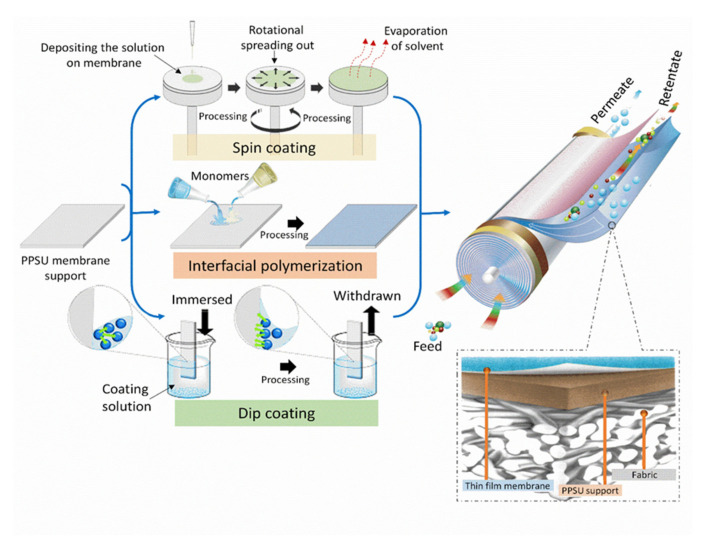
A schematic of the PPSU surface modifications using physical and chemical surface modification techniques.

**Figure 12 membranes-12-00247-f012:**
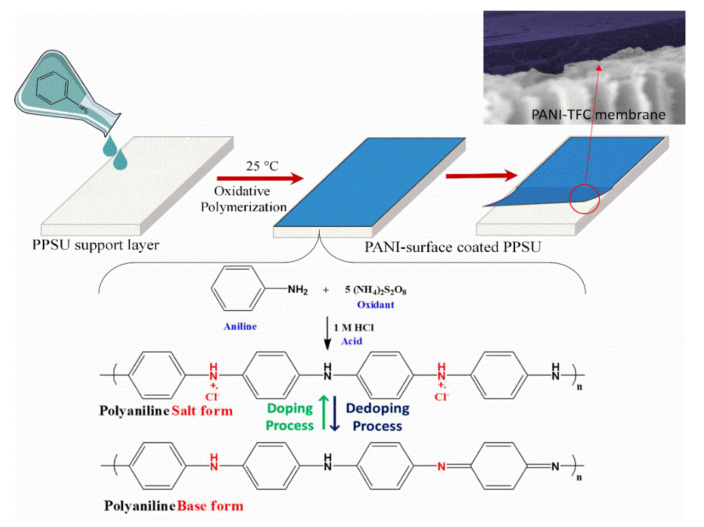
PANI-surface Polyphenylsulfone membrane surface coating via the chemical oxidation of monomeric aniline.

**Figure 13 membranes-12-00247-f013:**
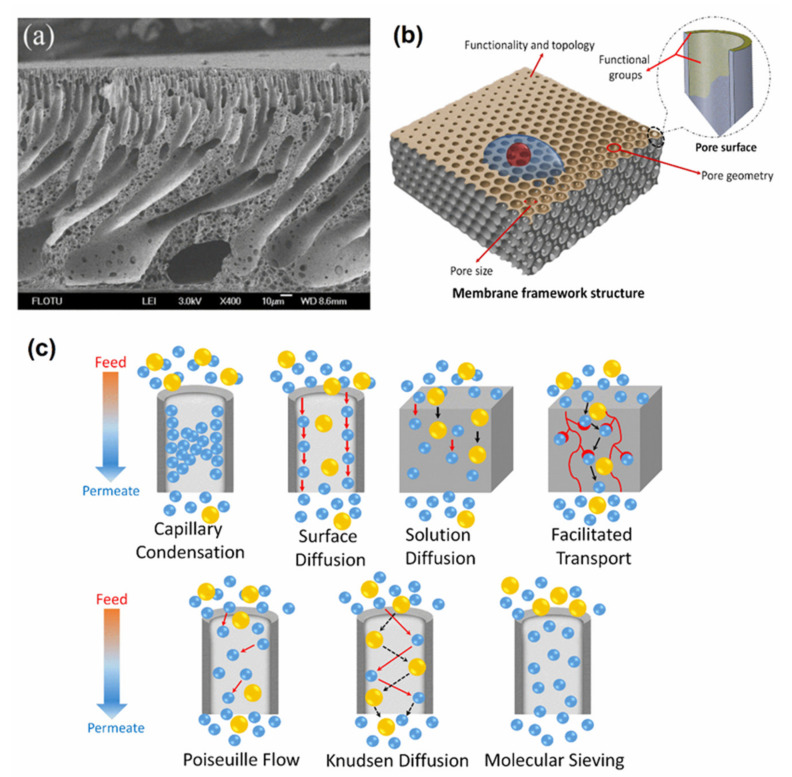
(**a**) SEM image of manufactured integrated asymmetric porous membrane; (**b**) schematic representation of pore size, pore geometry, pore surface, and structure of porous polymers; (**c**) schematic illustration of foremost transport mechanisms.

**Figure 14 membranes-12-00247-f014:**
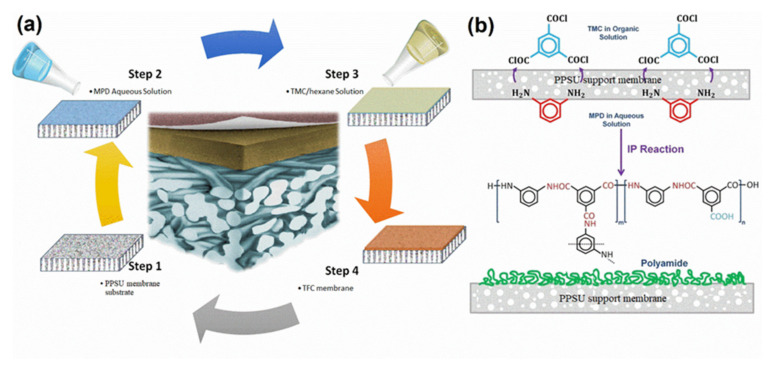
PPSU membrane surface modification schematic diagram: (**a**) steps in the modification process; (**b**) interfacial polymerization reaction mechanisms during the formation of the TFC layer.

**Table 1 membranes-12-00247-t001:** Summary of frequently used methods for modification of PPSU membrane and performance.

Description	Methods of Modification	Modifier Agents	Process of Membrane	Application	Performance	Ref.
Proton-conductive sPPSU membranes	Sulfonation	SO_3_ and (CH_3_)_3_ SiSO_3_Cl	Solvent evaporation	Electrochemical	(CH3)_3_SiClSO_3_ gave a homogeneous sPPSU with better control of the DS values as high as 1.0; asymmetric structure; high mechanical stability; proton conductivity about 55 mS/cm at 80 °C	[51]
Proton-conducting fuel cell sphPPSU membranes	Sulfophenylation	BuLi (metalating agent) and 2-sulfobenzoic acid cyclic anhydride	Vacuum dry	Fuel cells	sphPPSU showed DS values as 0.9; membranes have high thermal stability (300 and 350 °C); the proton conductivity about 60 mS/cm at 70 °C	[52]
PEI/PPSU sheet	Blending	PEI	Direct injection molding	Plasticization	PEI/PPSU blends are miscible; elasticity and yield stress changed linearly with PEI-rich blends composition	[53]
Proton exchange SPEEK/SiSPPSU membranes	Silylation and sulfonation; and blending	PhSiCl_3_ and H_2_SO_4_; SPEEK	Solvent evaporation	Fuel cells	SiSPPSU showed DS values as 2.0; exhibited high and stable conductivity values at 120 °C when dry (6.1 × 10^−3^ S/cm) and wet conditions (6.4 × 10^−2^ S/cm)	[54]
sPPSU-proton conducting membrane	Sulfonation	H_2_SO_4_ and ClSO_3_Si (CH_3_)_3_	Sol-gel processes	Fuel cells	sPPSU reached the conductivity values as high as 1.1 × 10^−2^ S cm^−1^ at 130 °C	[55]
PPSU/PBNPI membrane	Blending	PBNPI	Solvent evaporation	Hydrogen separation	The gases H_2_, CO_2_ and CH_4_ permeability increased up to 50%	[56]
PPSU/PBNPI membrane	Blending; immersion method	PBNPI; p-xylylenediamine (crosslinking reagent)	Solvent evaporation	Gas permeation	O_2_ and N_2_ permeation rates of 23.2 and 22.42	[57]
sPOSS/sPPSU composite proton exchange membranes	Blending	sPOSS	Dry	Fuel cells	sPOSS/sPPSU composites multilayered structure and reduce brittleness; conductivity 1 × 10^−2^ S cm^−1^ at 90 °C	[1]
Ionic exchange sPPSU/sPES membrane	Sulfonation; Blending	H_2_SO_4_; sPES	Solvent evaporation and dry	Fuel cells	The membrane surfaces show the smoother about 2 nm; stress–strain values 80 MPa and 7%	[5]
SPEEK/SiPPSU composite membranes	Silylation; Blending	SPEEK	Dry	Fuel cells	The presence of silicon enhances the temperature of loss of sulfonic acid groups; composites show superior behavior in terms of mechanical properties (higher elastic modulus and tensile strength)	[50]
PPSU/PEI membranes	Blending	PEI; PEG 200	Wet phase inversion	Ultrafiltration	Asymmetric and spongelike structure; water contact angle decreases significantly upto 64° and EWC 59.37%; IEP shifted pH 8 and shown positive charge; flux 545.54 kg m^−2^ h^−1^; rejection 56%	[20]
sPPSU positively charged membrane	UV grafting	[2-(methacryloyloxy)ethyl]trimethyl ammonium chloride; diallyldimethylammonium chloride		Nanofiltration; textile dyes	Spongelike morphology; MWCO 1627–1674 Da; PWP of 9–14 LMH bar^−1^; rejection of MgCl_2_ (95%) and Safranin O dye (99.9%)	[58]
PPSU thin-film composite membrane	Oxygen plasma (pretreatment); surface modification	2,5-bis(4-amino- 2-trifluoromethyl-phenoxy)benzenesulfonic acid; 4,4-bis(4-amino-2-trifluoromethyl-phenoxy)biphenyl-4,4-disulfonic acid	interfacial polymerization	Nanofiltration; dye removal	Water flux 63.9 and 71.3 L/m^2^ h; dye rejection 48–80%	[59]
sPPSU/sPES membranes	Sulfonation; Blending	H_2_SO_4_; sPES	Crosslinking; heat and dry	Fuel cells	Maximum conductivity of 0.12 S/cm	[60]
sPPSU TFC membranes	Surface modification	MPD;TMC	Interfacial polymerization	Forward osmosis	Water flux up to 54 LMH with 8.8 gMH salt reverse flux under PRO mode	[61]
PPSU/PI solvent resistant membrane	Blending	PI	Phase inversion; solvent evaporation	Nanofiltration	Asymmetric structure with a dense skin layer; highest flux for alcohol and alkanes was achieved for a 50/50 wt.% blend;	[62]
PPSU/TiO_2_ nanocomposites membrane	Blending	TiO_2_	Solvent evaporation	Biomedical	Nanocomposites shown active inhibition against *E. coli* and *S. aureus* bacteria with and without UV irradiation; the stiffness, strength, toughness, hardness and heat distortion temperature increases	[63]
Anion exchange PyPPSU membrane	Blending	1-methyl-2-pyrrolidone	Solvent evaporation	Vanadium redox flow battery	Vanadium ions permeability (0.07 × 10^−7^–0.15 × 10^−7^ cm^2^ min^−1^); coulombic efficiency of 97.8% and energy efficiency of 80.2%	[64]
PPSU solvent resistant membrane	Blending	Cu-BTC	Phase inversion	Nanofiltration; methanol–dye separation	Improve tensile strength 29%; methanol flux 135 L m^−2^ h^−1^	[65]
PPSU nanofibrous membrane	Blending	PEG 400	Electrospinning	Wastewater treatments	Water contact angle 8.9°; porosity 72.4%; water flux 7920 L/m^2^h	[66]
PPSU membranes	Blending	sPPSU	Phase inversion	Ultrafiltration	Porosity 48%; MWCO 70 kDa; pure water flux 218 L m^−2^ h^−1^; FRR 79%; BSA rejection 85%	[49]
sPPSU/PIM-1 membrane	Blending	sDCDPS; PIM-1	Slower solvent evaporation	Gas Separation	The tensile strength up to 72 MPa and extension at break 3.5%; the gas separation performance above the Robeson upper bounds for O_2_/N_2_, CO_2_/N_2_, CO_2_/CH_4_	[67]
PPSU/FAC composite membrane	Blending	FAC	Phase inversion	Phenol filtration	Fragmented surface and spongy porous linkages; contact angle 43.8°; porosity 30%; pure water flux 26 Lm^−2^ h^−1^, phenol rejection 96.4%	[68]
MgO/sPPSU/PPSU membranes	Blending	MgO; sPPSU	Phase inversion	Ultrafiltration; Oil separation	Porosity 65% and MWCO 70 kDa; contact angle 48°; FRR 85% and HA rejection 63% and castor oil rejection 99%	[69]
PPSU/Cu-BTC solvent resistant nanofiltration	Blending	Cu-BTC	Phase inversion	Nanofiltration; dye and methanol separation	Contact angle 61°, and porosity 62%; Flux 19 L/m^2^ h and rejection of methanol 93%	[70]
sPPSU proton exchange membrane	Sulfonation; Blending	H_2_SO_4_	Solvent evaporation	Fuel cells	Conductivity of 0.1 S/cm and power density of 471 mW/cm^2^ at 80 °C	[71]
PPSU membrane	Blending	PVP; PEG; Tween 80	Phase inversion	Ultrafiltration	Water flux 148 L/m^2^ h; BSA rejection increased from 53.2% to 81.5%	[30]
sPPSU asymmetric membranes	Sulfonation; Blending	TMSClS	Phase inversion	Ultrafiltration	Decomposition temperature at 510 °C; contact angle 33°, and porosity 51%; FRR 70%	[72]
sPPSU/f-SWCNTs mixed-matrix membranes	Sulfonation; Blending	3,3′-disulfonated 4,4′-dichlorodiphenyl sulfone; f-SWCNTs	Phase inversion	Gas separation	Enhanced the permeability for N_2_, O_2_, He, and CO_2_ and the selectivity for O_2_/N_2_ and O_2_/CO_2_	[73]
Porous PPSU membrane	Blending	Carrageenan	Phase inversion	Ultrafiltration	Contact angle 43° and porosity 78%; zeta potential −24 mV at pH 7; permeability increased up to 29 Lm^−2^ h^−1^ bar^−1^	[74]
PPSU/GO mixed matrix membrane	Blending	GO; PEG1000	Phase inversion	Ultrafiltration	Hydrophilicity and the thermal stability improved; pure water flux 132 L·m^−2^·h^−1^ and the rejection 96.8%	[28]
PPSU/Zeolite mixed matrix membrane	Blending	Fe-ZSM-5; Cu-ZSM-5	Phase inversion	Organic compounds removal	Surface roughness increased (Ra- 18.52 nm); zeta potential about −57.2 mV at pH 7; water flux of 62 L·m^−2^·h^−1^, lignin rejection up to 88.5%	[31]
PPSU/BiOCl-AC membrane	Blending	BiOCl-AC; PVP	Phase inversion	Ultrafiltration; oil separation	Asymmetric structures with thick top layer; contact angle 67°; pure water flux 465 L·m^−2^·h^−1^; rejection diesel fuel 80% and 90% of crude oil	[42]
Alkali resisting PPSU membrane	Blending	PVP- 10, 55, 360, and 1300 kDa	Phase inversion	Ultrafiltration	Asymmetric and fingerlike structure; Tensile strength upto 2.53 MPa for 10 kDa; MWCO ranged from 2 kDa to 175 kDa; pure water flux 69 L·m^−2^·h^−1^; better anti-alkali property in NaOH solution (pH = 13)	[13]
HBE–MMT/PPSU nanocomposite membrane	Blending	Functionalized montmorillonite	Phase inversion	Water treatment	Contact angle 53.6°; pure water flux about 380 L·m^−2^·h^−1^ at 5 bar; rejection of salt 40–50%	[75]
Polyamide TFN PPSU membrane	Blending; Surface modification	GO (support layer); PIP and TMC	Interfacial polymerization	Nanofiltration; l kinetic hydrate inhibitor (KHI) removal	KHI rejection of 99% and permeation flux of 32.7 L/m^2^ h (at 9 bar and feed concentration of 0.5 wt.% KHI)	[76]
sPPSU/TiO_2_ mixed matrix hollow fiber membranes	Blending	TiO_2_	Phase inversion	Ultrafiltration	Pure water flux 60 L·m^−2^·h^−1^; contact angle 67°; rejection of BSA 91%	[77]
PPSU membrane	Blending	PEG 400; PEG 20000	Phase inversion	Filtration of aqueous media	Porosity 72%; tensile Strength at Break 7.75 MPa and elongation at Break 50.14%; Pure water flux 19 L·m^−2^·h^−1^ (PEG400) and 183 L·m^−2^·h^−1^ (PEG20000); 100% turbidity rejection	[10]
PPSU membrane	Blending	PEG 400; PEG 2000; PEG 6000; PEG 20000; PEG 35000; PEG 40000	Phase inversion	Ultrafiltration	Contact angle 50° to 90°; pure water flux of 486 Lm^−2^ h^−1^; human serum albumin rejection 90%	[78]
Ionic crosslinked sPPSU membrane	Surface modification	HPEI	Coating	Nanofiltration; organic solvent filtration	Ethanol permeability 1.47 L m^−2^ h^−1^ bar^−1^; rejection of 99.9% to Rose Bengal dye	[79]
High-Flux PPSU membranes	Blending	PEG 6000–40000	Phase inversion	Ultrafiltration	Pure water flux 500–1000 L m^–2^ h^–1^ at 0.1 MPa; 90% rejection of human serum albumin (PEG20000)	[80]
PA-MOF/PPSU-GO TFN membrane	Blending; Surface modification	GO (support layer); MOF; PIP and TMC	Interfacial polymerization	Nanofiltration	Permeate flux 59.9 L/m^2^·h; KHI rejection 96%; FRR 97.8% and an excellent long-term stability	[81]
sPPSU/PBI membrane	Blending; crosslinking	PBI; DBX (crosslinker)	Heat and solvent evaporation	Nanofiltration; organic solvent removal	Permeability 11.8 Lm^−2^ h^−1^ bar^−1^; rejection of tetracycline 97%.	[82]
Double crosslinked sPPSU/PBI membrane	Blending; crosslinking	PBI; DBX (crosslinker)	Heat and solvent evaporation	Nanofiltration; hydrogen purification	H_2_ permeability of 46.2 Barrer and a high H_2_/CO_2_ selectivity of 9.9 at 150 °C	[83]
Amine functionalized PPSU membrane	Amination; Blending	SnCl_2_; HNO_3_	Phase inversion	Nanofiltration; dye removal	Pore size of 0.72 nm; positively charged active layers; contact angles 31°; pure water flux ∼54 Lm^−2^ h^−1^; CaCl_2_ and AlCl_3_ multivalent salts rejection 89% and 93.5%; crystal violet dye rejection > 99%	[84]
High-performance PPSU/sPANI membrane	Blending	sPANI	Nonsolvent induced phase separation	Ultrafiltration	Contact angle was 57°; porosity 81%; BSA adsorption value of 3.6 μg/cm^2^; water flux of 260 L/m^2^ h; BSA rejection 95%	[40]
PPSU/carboxylated GO nanocomposite membrane	Blending	Carboxylated GO	Phase inversion	Nanofiltration; heavy metal removal	Surface charge of −70 mV; flux of 27 L m^−2^ h^−1^; rejection of As(V) 96%, Cr(VI) 93%, Zn^2+^(81%), Cd^2+^ (74%), Pb^2+^ (73%)	[85]
sPPSU membrane	Sulfonation	H_2_SO_4_	Phase inversion	Ultrafiltration; heavy metal and protein separation	Water flux of 190.33 Lm^−2^ h^−1^ and FRR of 86.56%; protein rejection of 66.3%, 74.0% and 91.2% for trypsin, pepsin, and BSA; Cd^2+^ and Pb^2+^ ions rejection of 75.2% and 87.6%;	[86]
PPSU/carboxylated GO nanocomposite membrane	Blending	Carboxylated GO	Phase inversion	Ultrafiltration; Antimicrobial and antifouling	Bacteriostasis rates of 74.2%,81.1% and 41.9% against *E. coli*, *P. aeruginosa* and *S. aureus*; FRR 95.3%	[87]
Porous PPSU/sPEEK membrane	Blending	sPEEK	Solvent evaporation	Vanadium flow batteries	Contact angle 47°; tensile strength 2.78 MPa; proton conductivity of 14.3 mS cm^−1^ at 15 °C	[88]
PPSU/SnO_2_ mixed matrix hollow fiber membrane	Blending	SnO_2_	Vacuum evaporation	Ultrafiltration; dyes removal	Contact angle 63°; porosity 84%; pure water flux 362.9 L/m^2^ h; dyes rejection about >94% for RB-5, and >73% for RO-16	[89]
PPSU/CuO/g-C_3_N_4_ membrane	Blending	CuO/g-C_3_N_4_	Nonsolvent induced phase inversion	Ultrafiltration; antifouling and protein separation	Smooth surfaces Ra-9.8 nm; increase pores on the top layer as well as in the sublayer; contact angle 48°; water flux 202 L/m^2^h; BSA protein rejection 96%; FRR 79%	[90]
Super-hydrophilic PPSU TFC membrane	Surface modification	MPD and TMC	Electrospun; plasma treatments; interfacial polymerization	Forward osmosis	Contact angle 0°; Osmotic water flux 14 L/m^2^h	[91]
PPSU hollow fiber membranes	Blending	CA; CAP	Dry-wet spinning	Ultrafiltration; arsenic removal	Contact angle 60° and 43°; arsenic removal 34% and 41%; pure water permeability 61.47 L/m^2^h bar and 69.60 L/m^2^ h bar; FRR 88.67%	[92]
PPSU/silver-hydroxyapatite nanocomposite membrane	Blending	silver-hydroxyapatite	Phase inversion	Ultrafiltration; organic matter removal	Porous and honeycomblike structure; contact angle 60°; rejection 89%	[93]
Proton exchange sulfonated PPSU/PSU membrane	Sulfonation	Trimethylsilyl chlorosulfonate;	Vacuum dry	Fuel cells	Proton conductivity 34.1 mS cm^−1^ at 70 °C; power density of 400 mW cm^−2^; current density of 1100 mA cm^−2^	[35]
PPSU/Ag-MWCNTs nanocomposite membrane	Blending	Ag-MWCNTs	Phase inversion	Nanofiltration; ion removal and antibacterial activity	Zeta potential −78 mV; contact angle 49°; porosity 73%; rejection of Na_2_HAsO_4_ 99.5% and Na_2_Cr_2_O_7_ 100%	[87]
PPSU/MWCNTs membrane	Blending	MWCNTs	Phase inversion	Ultrafiltration; heavy metals removal	Dense skin layer on top and a porous supportive sub-layer; surface roughness Ra 21 nm; contact angle 61°; porosity 50%; flux 186 L/m^2^ h rejection of Pb^2+^ (>98%), Hg^2+^ (>76%) and Cd^2+^ (>72%)	[94]
PPSU/ZnO nanocomposite membrane	Blending	ZnO	Phase inversion	Nanostructured- hybrid membranes; anionic dye; antimicrobial; wastewater treatment	Pore size 0.75 nm; zeta potential –65.7 mV at pH 7; methyl orange dye rejection 98% with a water flux 19 L/m^2^h; antibacterial activity of *E. coli* (6.2) and *S. aureus* (6.8)	[95]
Hydrophilic PPSU membranes	Blending	1,2-propandiol; PVP	Nonsolvent induced phase separation	Ultrafiltration	Contact angles of 46.4°;Water flux 674 kg m^−2^ bar^−1^h^−1^	[96]
PPSU/PES/SiO_2_ nanocomposite membrane	Blending	PES; SiO_2_	Vapor induced phase separation; nonsolvent induced phase separation	Ultrafiltration	Water flux 76.65 L/m^2^·h; BSA retention of 82.01%;	[97]
Silica filled PPSU/PDMS Composite Membranes	Surface modification	PDMS; Silica	Coating	Biobutanol Separation	Weight loss starts from 400 °C; contact angle ∼130°; flux 536 g. m^−2^ h^−1^; butanol separation factor 30.6	[36]
PPSU/PANI hollow fiber membrane	Blending	PANI	Dry-jet wet spinning	Humic acid removal	Zeta potential −16 mV at pH 9; Water flux 127 L/m^2^h; Humic acid rejection 98%;	[98]
Proton exchange sPPSU membrane	Sulfonation	H_2_SO_4_; CNDs (crosslinker)	Vacuum dry	Fuel cells	Proton conductivity 10^−2^ S/cm at 120 °C.	[99]
PPSU/Al-MOF mixed matrix membrane	Blending	Al-MOF	Phase inversion	Ultrafiltration,; dye separation; antifouling	Contact angle 63°; surface roughness Ra 21.9 nm; pure water flux 47 L·m^−2^·h^−1^; FRR 93%; rejection of organic dye methyl violet 93.8%	[100]
PPSU/CA/ZrO_2_ hollow fiber membranes	Blending	CA; ZrO_2_	Dry-wet spinning	Arsenic Removal	Surface roughness Ra 43 nm; contact angle 48°; permeability of 89.94 L/m^2^h bar; removal of arsenic 87%	[45]
PPSU/CA hollow fiber membrane	Blending	CA	Dry–wet spinning	Removal of dyes	Permeability 64.47 L/m^2^ h bar; removal of Reactive black 5 dye 95%	[101]
PPSU/Zn-MOF composite membrane	Blending	Zn-MOF	Phase inversion	Ultrafiltration; antifouling	Asymmetric structure and dense microporous active skin layer; surface roughness Ra 13.88 nm; porosity 72%; tensile strength 7.9 MPa; permeability 33 L m^−2^ h^−1^ bar^−1^; FRR 98%	[102]
PPSU/CA/ZnO-MgO hollow fiber membrane	Blending	CA; ZnO-MgO	Dry–wet phase inversion	Arsenic removal	contact angle 60°; permeability 69.58 L/m^2^h bar; arsenic rejection 81.31%; FRR 91%	[103]
PANI coated PPSU Membranes	Surface modification	PANI	Coating	Dye separation; antibacterial activities	Surface roughness Ra-3.15 nm; contact angle 55°; zeta potential −1.7 mV at pH 6; permeability 53 L·m^−2^·h^−1^·bar^−1^; rejection of methylene blue dye 96%; bacteriostasis of *E. coli* 95% and *S. aureus* 88%	[104]

## Data Availability

Not applicable.

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
