# Peer review of "Recent Advancements in Polyphenylsulfone Membrane Modification Methods for Separation Applications"

_membranes, 2022, doi:10.3390/membranes12020247_

Round 1

Reviewer 1 Report

The article Recent advancements in polyphenylsulfone membrane modification methods for separation applications is valuable and suitable for publication in Membranes.

My main point is the amount of information and volume. I understand that it is a review but in this form it is more suitable as a chapter of a book on membrane technology?

I believe that my detailed comments to the authors will be useful.

Introduction

Line 45

susceptible instead vulnerable

 I suggest to give some more information in Introduction chapter about membrane fouling problem

“academic and industrial researchers”- what is the difference?

Figure 2- just remove from your figure term “etc.

“The article starts with a brief introduction to PPSU polymer” –there is no need to write such information.

I suggest to write in the end of chapter (after figure 2) information what was the reason to prepare this review and for whom. You gave it earlier but it will be more convenient to write it clear in the end of chapter Introduction.

Chapter 2

good chemical resistance- this is not clear, maybe you can describe it a little bit more (resistance).

Table 1 is very important. Just check such terms as : SO3 not SO3, H2SO4 not H2SO4………etc.

All Latin names are written in italics

Is it possible to decrease quantity of information in Performance column? Table will be more convenient for readers.

Figure 6 is not clear for me. Also figure 9 can be improved. Maybe you can limit the amount of information in this figure? It looks like a graphic abstract.

Figure 7 can be improved –especially resolution of “tensile straight” Maybe it will be better to create two figures instead one.

Conclusions and Future Prospects  chapter: In my opinion you can improve it

I suggest presentation of selected specific applications in points - a lot of information is in the text of the article, perhaps here it is necessary to really briefly describe in hash words what is most important in the analyzed technique since 2002.

The future development can also be described briefly.

The amount of literature cited by the authors confirms the nature of the publication

Author Response

Query (1): Line 45, susceptible instead vulnerable

Response: We thank the reviewer for the accurate summary of our work, especially for positive comments on the originality and quality of our research work and appreciation of the importance of the findings. As per the reviewer's suggestion, in the revised manuscript, the author's replaced the “susceptible instead vulnerable” from the text.

Query (2): I suggest to give some more information in Introduction chapter about membrane fouling problem

Response: We thank the reviewer's suggestion, in the revised manuscript, we added some more information in the Introduction chapter about the membrane fouling problem.

Query (3): “academic and industrial researchers”- what is the difference?

Response: The reviewer has pointed out the difference between “academic and industrial researchers”. Therefore, according to our knowledge “Academia” is famous for its emphasis on research and discovery of new materials, and much research is conducted for the purpose of learning. In contrast, “industry” work allows researchers to feel a sense of immediate impact on consumers.

Query (4): Figure 2- just remove from your figure term “etc.

Response: The Reviewer has correctly pointed out Figure 2, in the revised manuscript we removed from the figure the term “etc.

Query (5): “The article starts with a brief introduction to PPSU polymer” –there is no need to write such information.

Response:  We completely agree with the reviewer's observation about “The article starts with a brief introduction to PPSU polymer”. In the revised manuscript we removed information from the text.

Query (6): I suggest writing at the end of the chapter (after figure 2) information what was the reason to prepare this review and for whom. You gave it earlier but it will be more convenient to write it clear in the end of chapter Introduction.

Response:  The authors would like to express their gratitude to the reviewer for taking an interest in this topic. As per the reviewer's suggestion, we write at the end of chapter Introduction in the revised manuscript.

Query (7): good chemical resistance- this is not clear, maybe you can describe it a little bit more (resistance).

Response:  As per the reviewer's suggestion, in the revised manuscript we described the chemical resistance in the text.

Query (8): Table 1 is very important. Just check such terms as : SO3 not SO3, H2SO4 not H2SO4………etc.

Response:  Thank you for taking such good care of our manuscript. In the revised manuscript, we have edited in Table 1.

Query (9): All Latin names are written in italics.

 Response:  As per the reviewer suggestion, in the revised manuscript we edit the Latin names in italics.

Query (10): Is it possible to decrease quantity of information in Performance column? Table will be more convenient for readers.

Response:  Thank you very much. It is a nice suggestion. We appreciate the reviewer's comment. However, there will be a huge shift in references sequences. In the future, we keep these valuable comments in mind for more convenience for readers.

Query (11): Figure 6 is not clear for me. Also figure 9 can be improved. Maybe you can limit the amount of information in this figure? It looks like a graphic abstract.

Response:  The quality of the Figures was improved in the revised manuscript.

Query (12): Figure 7 can be improved –especially resolution of “tensile straight” Maybe it will be better to create two figures instead of one.

Response:  We acknowledge that the resolution of Figure 7 is not clear. In the revised manuscript we increased the quality of the figure so that readers could understand them better.

Query (13): Conclusions and Future Prospects chapter: In my opinion, you can improve it

Response:  Thank you for taking such great care of our manuscript. In the revised manuscript, the conclusions and future prospects chapter was improved.

Query (14): I suggest presentation of selected specific applications in points - a lot of information is in the text of the article, perhaps here it is necessary to really briefly describe in hash words what is most important in the analyzed technique since 2002.

Response:  PPSU is an emerging candidate in sulfone family polymers. Most of the studies have been done between 2002 and 2021. In this period, blending PPSU with other suitable polymers is done, with the main aim of improving membrane porosity and water flux. On the whole, the formation of polymer composites and blends is more dominant in the literature, and it is already included in the text.

Query (15): The future development can also be described briefly.

Response:  We described briefly the future development in the main text of the revised manuscript.

Query (16): The amount of literature cited by the authors confirms the nature of the publication

Response:  The authors appreciate the reviewer's thoughtful comments. The majority of the papers cited are from SCOPUS, and they are not open access. They contain high-quality work and are published in peer-reviewed journals.

Reviewer 2 Report

In this review paper, various surface modification methods for polyphenylsulfone (PPSU) membranes and processes along with their mechanisms and performance are considered.  Results of this paper has important application in the field of application of membrane for separation applications.  Authors may wish to consider the following in revision of their manuscript.

  1. Please comment on the limitations of 3 main modifications reviewed in the paper.
  2. Please comment on cost of various modifications of PPSU membranes.
  3. Please include results of long term operations of membranes in the review paper.
  4. Please summarize math models of modified PPSU membranes in a table.
  5. Please summarize important operating parameters with values in a table for modified PPSU membranes.

Author Response

Query (1): Please comment on the limitations of 3 main modifications reviewed in the paper.

Response:  We thank the reviewer for the accurate summary of our work, especially for positive comments on the originality and quality of our research work and appreciation of the significance of the findings.

Introducing functional groups into the main PPSU polymer chain is a very complicated process, particularly when washing the polymer. There is also a low yield of functional group attachment.

Grafting is a multi-stage process.

Surface coating is also a challenge with PPSU polymers. That is why researchers are focused on the development of polymer composites and blends.

Query (2): Please comment on cost of various modifications of PPSU membranes.

Response:  It is nice to comment, however, researchers tend to choose "easy of process and low cost," with the formation of polymer composites and blends having low-cost techniques.

Query (3): Please include results of long term operations of membranes in the review paper.

Response:  Thank you for taking care of our manuscript deeply. The majority of PPSU-based membrane research has been done, and several parameters have been reported in the articles.

Query (4): Please summarize math models of modified PPSU membranes in a table.

Response:  Good suggestion, however, how to use mathematical models in the paper is new to us.

We will practice it and write it down in the future.

Query (5): Please summarize important operating parameters with values in a table for modified PPSU membranes.

Response:  Thank you very much for your comment; the membrane figure-of-merits are described in the table.

Reviewer 3 Report

Shukla et al. presented an interesting review on the ongoing efforts to modify polyphenlysulfone membranes for separation applications. This material is significant from the membrane technology point of view. The review discusses the various methods used for membrane modifications, including physical and chemical. The associated challenges are also discussed. The structure of the manuscript is logical. Therefore, the manuscript can be accepted for publication after the authors address the following comments:

  1. The problem of nanoparticle agglomeration is quite common in the case of the blending approach. Therefore, the authors should briefly discuss its status in the case of the PPSU membranes.
  2. Nowadays, anti-biofouling and antiviral properties have also gained significant attention among membrane researchers in the current scenario. Therefore, a short discussion on these properties would add value to the review.
  3. The status of the application of the PPSU membrane as a support membrane for preparing NF/RO membranes can also be discussed briefly.
  4. The authors should cite the source article for the data presented in the supplementary file.
  5. Provide good quality images for Figure S3.
  6. Proofreading of the manuscript is required. Some work on improving English expression is also needed.

Author Response

Query (1): The problem of nanoparticle agglomeration is quite common in the case of the blending approach. Therefore, the authors should briefly discuss its status in the case of the PPSU membranes.

Response: Thank you for taking such good care of our manuscript. In the revised manuscript, the issue of nanoparticle agglomeration was briefly added in the main text in the case of the blending approach.

Query (2): Nowadays, anti-biofouling and antiviral properties have also gained significant attention among membrane researchers in the current scenario. Therefore, a short discussion on these properties would add value to the review.

Response: Thank you very much for the nice and valuable suggestion. In the revised manuscript, we have discussed the anti-biofouling and antiviral properties in the text.

Query (3): The status of the application of the PPSU membrane as a support membrane for preparing NF/RO membranes can also be discussed briefly.

Response: As per the reviewer's suggestion, in the revised manuscript, we discussed briefly the status of the application of the PPSU membrane as a support membrane for preparing NF/RO membranes in the main text.

Query (4): The authors should cite the source article for the data presented in the supplementary file.

Response: In the revised manuscript, we cited the source article for the data presented in the supplementary file.

Query (5): Provide good quality images for Figure S3.

Response: We acknowledge that Figure S3 has a low resolution. We improved the quality of the figures in the revised manuscript so that readers could better understand them.

Query (6): Proofreading of the manuscript is required. Some work on improving English expression is also needed.

Response: Thank you for taking such great care of our manuscript. As per the reviewer's suggestion, the manuscript has been edited for the English language and spelling by “Enago”.